# A unified mechanism for the control of *Drosophila* wing growth by the morphogens Decapentaplegic and Wingless

**Myriam Zecca** [1,2], **Gary Struhl** [1,2]*

**1** Department of Genetics and Development, Columbia University, New York, New York, United States of America, **2** The Mortimer B. Zuckerman Mind Brain Behavior Institute, Columbia University, New York, New York, United States of America

* gs20@columbia.edu

**Data Availability Statement:** All relevant data are within the paper and its Support Information files.

**Funding:** The authors have received funding from the following: NIH (R01 GM113000 R35 GM127141) and the Howard Hughes Medical

## Abstract

Development of the *Drosophila* wing—a paradigm of organ development—is governed by 2 morphogens, Decapentaplegic (Dpp, a BMP) and Wingless (Wg, a Wnt). Both proteins are produced by defined subpopulations of cells and spread outwards, forming gradients that control gene expression and cell pattern as a function of concentration. They also control growth, but how is unknown. Most studies have focused on Dpp and yielded disparate models in which cells throughout the wing grow at similar rates in response to the grade or temporal change in Dpp concentration or to the different amounts of Dpp "equalized" by molecular or mechanical feedbacks. In contrast, a model for Wg posits that growth is governed by a progressive expansion in morphogen range, via a mechanism in which a minimum threshold of Wg sustains the growth of cells within the wing and recruits surrounding "pre-wing" cells to grow and enter the wing. This mechanism depends on the capacity of Wg to fuel the autoregulation of *vestigial* (*vg*)—the selector gene that specifies the wing state—both to sustain *vg* expression in wing cells and by a feed-forward (FF) circuit of Fat (Ft)/ Dachsous (Ds) protocadherin signaling to induce *vg* expression in neighboring pre-wing cells. Here, we have subjected Dpp to the same experimental tests used to elucidate the Wg model and find that it behaves indistinguishably. Hence, we posit that both morphogens act together, via a common mechanism, to control wing growth as a function of morphogen range.

## Introduction

Morphogens are "form-producing" substances that are made by distinct subpopulations of cells and move through surrounding tissue, decreasing in concentration as a function of distance from the source [1]. The resulting gradients thus confer spatial information, the amount of morphogen each cell receives providing an indication of its position relative to the source [2–4]. In several, well-established systems, morphogen gradients have been shown to generate a series of distinct concentration thresholds that organize the patterns of gene expression and

Institute (GS, Invesitigatorship) to GS. The funders had no role in study, design, data collection and analysis, decision to publish, or preparation of the manuscript.

**Competing interests:** The authors have declared that no competing interests exist.

**Abbreviations:** A/P, anterior–posterior; ap, apterous; BE, Boundary enhancer; D, Dachs; D/V, dorso–ventral; Dpp, Decapentaplegic; Ds, Dachsous; FF, feed-forward; Fj, Four-jointed; FRT, >, Flp recombinase targets; Ft, Fat; Hh, Hedgehog; Hth, Homothorax; IR, inner ring; ᵐTrap, Morphotrap; N, Notch; OR, Outer ring; QE, Quadrant enhancer; Tkv, Thickveins; Tsh, Teashirt; Tub, Tubulinα1; Vg, Vestigial; Wg, Wingless; Wg^Nrt, Neurotactin-Wingless; Wts, Warts; Yki, Yorkie.

cell fate (reviewed in [5,6]). Morphogens also govern organ growth. However, unlike patterning, we do not know how.

Here, we address this question in the *Drosophila* wing where growth depends critically on 2 morphogens, Decapentaplegic (Dpp, a BMP) and Wingless (Wg, a Wnt). Although Dpp and Wg belong to fundamentally different classes of signaling molecules with nonoverlapping mechanisms of signal production, reception, and transduction [7,8], they share a common logic of deployment and action in the developing wing (reviewed in [5,9]). Both are induced by short-range signaling between cells of abutting developmental compartments [10–15]. Both also accumulate as long-range gradients that extend away from these boundaries and set distinct threshold concentrations that control target gene expression and cell patterning [16–19]. Finally, both control wing growth. No growth occurs if either is absent or unable to signal [18,20–24]. Conversely, both induce supernumerary wing tissue when ectopically expressed [10,11,16–19,25].

Strikingly, although the patterns of gene expression and cytodifferentiation depend on the graded distributions of Dpp and Wg, no such relationship is observed for growth. Instead, cells throughout the prospective wing appear to gain mass and proliferate at similar rate regardless of their position [26–28]. This key observation poses the question of whether, and if so how, growth is governed by the amount of morphogen cells received.

Most studies have focused on Dpp as the quintessential morphogen and have given rise to many different hypotheses (reviewed in [29–35]). Prominent among them are models in which individual wing cells grow in response to the slope of the Dpp gradient [36–39] or to the change in the amount of Dpp they receive over time [40–42]. Other models retain the notion that wing cells grow in response to the absolute amount of Dpp they receive, but invoke additional feedback mechanisms that enhance the response to low levels and/or weaken the response to high levels. Many different kinds of feedback have been proposed, e.g., involving the modulation of extracellular signals (reviewed in [43–46]), changes in signal transducing capacity [43,47–56], or the differential buildup of mechanical tension [57–61]. Yet another class of models proposes that early exposure of wing cells to morphogen initiates a "cellular memory" that sustains their growth in the absence of subsequent signaling [13,25,62,63]. Thus, despite the uniquely powerful approaches for analyzing development in *Drosophila*, the question of how Dpp controls wing growth remains unresolved.

In previous work on Wg, we elucidated an alternative model, potentially applicable to Dpp, in which wing growth is governed by the progressive outward expansion in morphogen range [64–66]. Critical to this model was our demonstration of 2 distinct mechanisms of growth, both fueled by Wg through the control of the selector gene *vestigial* (*vg*), a transcription factor that defines the wing state [67–69]. The first is a recruitment mechanism by which the spread of Wg from cells flanking the dorso–ventral (D/V) compartment boundary induces surrounding "pre-wing" cells to grow, initiate *vg* expression, and enter the wing. The second is a maintenance mechanism in which Wg is required to sustain *vg* expression as well as the growth of wing cells behind the recruitment interface. Thus, as Wg spreads outwards from producing cells, it delineates a continuously expanding domain in which Wg levels exceed a minimum threshold necessary to recruit pre-wing cells and sustain their subsequent growth as wing cells.

Importantly, both the recruitment and maintenance mechanisms depend on the capacity of Wg to fuel the transcriptional autoregulation of *vg*. For the recruitment mechanism, the autoregulation is indirect and intercellular: Vg activity in wing cells programs them to send a "feed-forward" (FF) signal, the protocadherin Fat (Ft), that acts with Wg to induce *vg* expression in neighboring pre-wing cells [66]. The newly recruited wing cells then serve as a new source of Ft signal, propagating recruitment to the next tier of pre-wing cells in response to the outward spread of Wg. For the maintenance mechanism, the autoregulation is direct and

intracellular: Vg bound to the *vg* locus [69,70] autoregulates its own expression in response to Wg. Thus, as Wg spreads outwards from D/V border cells, it operates through both autoregulatory circuits to recruit and then sustain a continuously expanding population of wing cells [64–66].

Here, we ask if the FF model also applies to Dpp. To do so, we have subjected Dpp to the same experimental tests on which the model for Wg is predicated. We find that by every such test, Dpp behaves indistinguishably from Wg: It is critically required to fuel the FF recruitment and subsequent growth of wing cells, and it appears to do so by driving both the inter- and intracellular circuits of *vg* autoregulation. Finally, we have performed parallel experiments that confirm that both morphogens control wing growth as a function of their capacities to spread. Our results thus consolidate a unified mechanism for the control of wing size by morphogen range.

## Results

### The control of wing growth by Wg

Our primary goal is to test if Dpp controls wing growth by the same FF signaling paradigm we elucidated for Wg [64–66]. Accordingly, we first present the paradigm as it applies to Wg (Fig 1) and summarize the experimental criteria on which it is based. In subsequent sections, we apply these same criteria to test the role of Dpp.

**The FF paradigm.** Early in larval life, the nascent wing imaginal disc is partitioned into complementary body wall and "pre-wing" territories (Fig 1B) that correlate with the presence or absence, respectively, of 2 transcription factors Teashirt (Tsh) and Homothorax (Hth) [77–80]. The FF paradigm operates only in the pre-wing, Tsh$^{OFF}$ Hth$^{OFF}$ region [65]. Accordingly, we focus exclusively on analyzing the consequences of manipulating Dpp, Wg, and FF signaling in this domain.

Although the founding cells of the wing disc are first defined in the embryo by activation of the wing selector gene *vg* [67,81], this early *vg* expression subsides by the middle of the second larval instar (Fig 1A and 1B). Hence, the continued specification and growth of the wing depends on the reinitiation and subsequent maintenance of *vg* transcription in prospective wing cells within the pre-wing territory (Fig 1C and 1D). This occurs by 2 distinct mechanisms mediated by separate *cis*-acting "Boundary" and "Quadrant" enhancers in the *vg* locus (BE and QE; Fig 1E) [68,69,82].

First, the entire wing disc is segregated into abutting dorsal (D) and ventral (V) compartments by activation of the selector gene *apterous* (*ap*) in the founding cells of the D compartment [83–87]. This results in bidirectional DSL/Notch signaling across the D/V compartment boundary that induces "border" cells on both sides to express *vg* via the BE (Fig 1C and 1E "Border" cell) [68,69,82]. Importantly, Notch activation also induces D/V border cells within the pre-wing territory to express Wg [13–15], thus creating a "seed" population of *vg*-expressing, Wg-secreting wing cells within the pre-wing domain (Fig 1C).

Second, Wg secreted by D/V border cells fuels 2 modes of Vg autoregulation, both governed by the QE (Fig 1D and 1E) [64–66]. In the first mode, D/V border cells, by virtue of their expression of Vg, are programmed to proliferate in response to Wg. However, as this population of Vg-expressing cells expands, cells at the dorsal and ventral edges of the growing wing are displaced from the D/V boundary, lose access to DSL/Notch signaling, and switch to relying on Wg to sustain Vg expression and to continue proliferating as wing tissue. They do so by engaging an intracellular circuit of Vg autoregulation in which Vg, initially expressed under BE control, acts directly on the QE to maintain its own expression in response to Wg (Fig 1E, "Wing" cell). In the second mode, Vg activity programs wing cells to send a short-

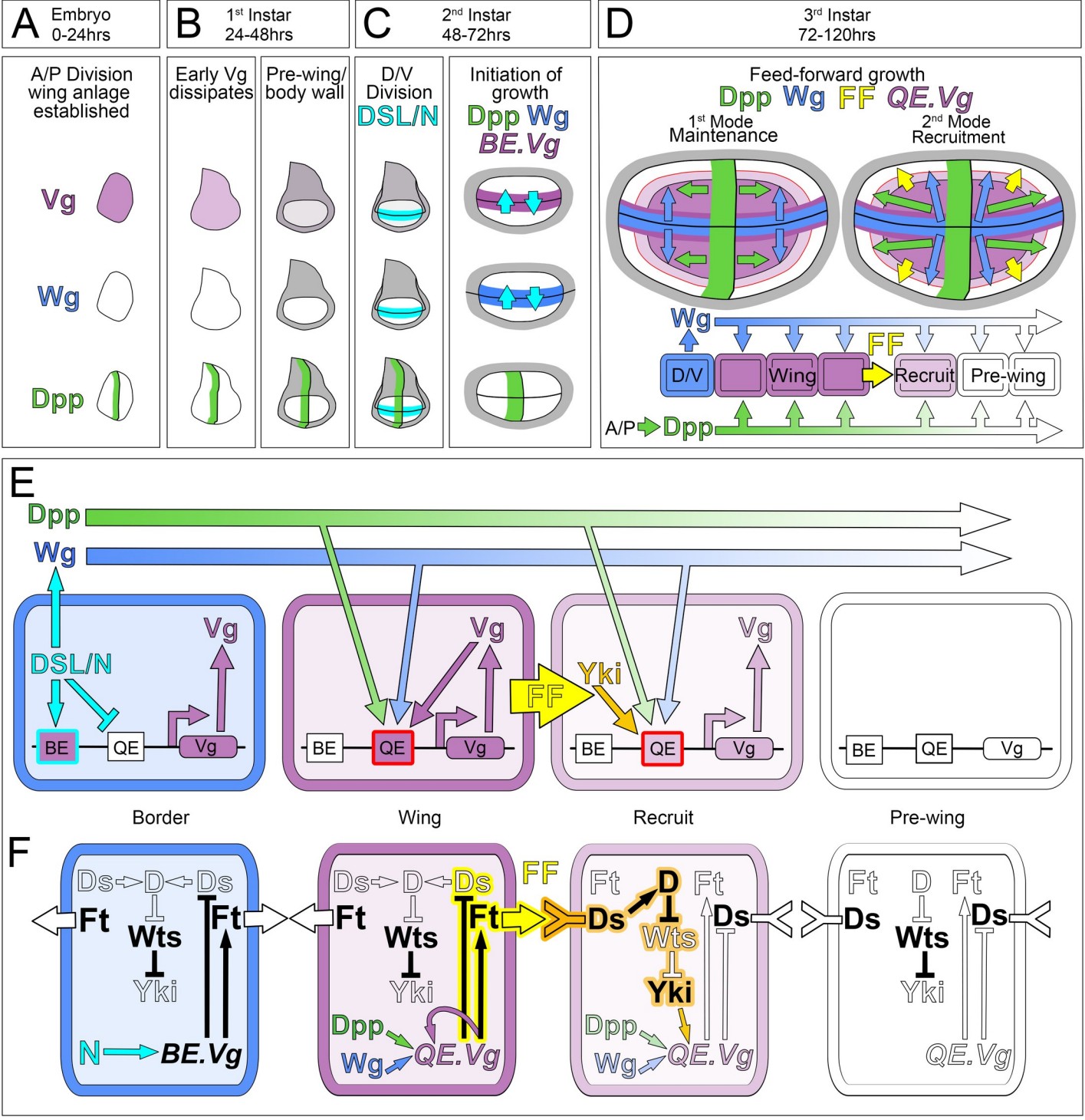

**Fig 1. FF model for the control of wing growth by Dpp and Wg. (A–C)** Initiation of FF wing growth. The establishment of the embryonic wing primordium (**A**), segregation of the pre-wing (white) and body wall (gray) territories during the first larval instar (**B**), and the segregation of the D/V compartments during the second larval instar (**C**). Dpp (green) is produced by a stripe of A cells along the A/P compartment boundary from mid-embryogenesis onwards. DSL/Notch signaling across the nascent D/V compartment boundary (turquoise) induces cells on either side to express Vg (purple) and Wg (blue), which act in conjunction with Dpp to initiate and sustain wing growth during the remainder of larval life. (**D**) Two modes of FF growth. **First Mode: Maintenance**: Cells specified as wing by Vg are programmed to grow in response to both Dpp and Wg; the continued expression of Vg (and hence maintenance of the wing state) depends on Vg sustaining its own expression in response to Dpp and Wg. **Second Mode: Recruitment**: Wing cells produce a *vg*-dependent, *vg*-inducing FF signal (yellow) that acts together with Dpp and Wg to induce neighboring pre-wing cells to grow, initiate Vg expression and enter the wing. Upon entry these cells switch to the maintenance mode and serve as a new source

of FF signal, propagating the continuing growth and recruitment of pre-wing cells in response to Dpp and Wg. Accordingly, the wing comprises 3 populations of Vg-expressing cells programmed to gain mass and proliferate in response to minimum thresholds of Dpp and Wg: (i) cells in the vicinity of the D/V border, which express Vg under DSL/Notch control; (ii) cells that descend from D/V border cells that proliferated away from the D/V boundary and switched to expressing Vg under Dpp and Wg control; and (iii) cells that descend from pre-wing cells induced by FF signaling, Dpp and Wg to initiate Vg expression and enter the wing; the latter 2 populations differ only by their provenance. **(E)** Vg regulatory circuits governing FF wing growth. Vg expression in the developing wing depends on distinct "Boundary" and "Quadrant" enhancers (BE and QE; arranged in this depiction by the logic of their deployment rather than by their normal position in the *vg* locus). **Border**: Notch signaling induces Wg expression and acts via the BE to initiate Vg expression in D/V border cells while also blocking activity of the QE. **Wing**: As Vg-expressing D/V border cells proliferate in response to the outward spread of Dpp and Wg, they are displaced from the D/V boundary, lose Notch input, and switch to relying on the QE to sustain an intracellular circuit of Vg autoregulation fueled by Dpp and Wg. **Recruit**: Vg activity in wing cells programs them to send the FF signal (FF), which induces neighboring pre-wing cells to grow and to up-regulate nuclear Yki (orange). Yki is a transcriptional activator that forms a complex with the site-specific DNA-binding protein Scalloped (Sd) [71–73] to bind the QE to initiate Vg expression in response to Dpp and Wg. Like Yki, Vg is a transcriptional coactivator that can associate with Sd [70]. Hence, as the level of Vg rises, it substitutes for Yki to generate a Vg-Sd complex that binds directly to the QE to sustain its own expression in response to Dpp and Wg, recruiting the pre-wing cell into the wing. **Pre-wing**: The QE is inactive in cells positioned ahead of the FF recruitment interface owing to inadequate Dpp, Wg, and/or FF input. **(F)** The FF signaling pathway. The FF signal is the protocadherin Ft which is transduced by the protocadherin Ds. Ft and Ds function as bidirectional ligands and receptors for each other in PCP [74]. However, in the context of the developing wing, the "on" and "off" states of Vg in wing versus pre-wing cells biases their activities to generate a unidirectional Ft (wing) to Ds (pre-wing) signal in this context [66]. Specifically, Vg activity in wing cells drives expression of the ectokinase Fj [75] to promote the signaling activity of the protocadherin Ft [76] while at the same time repressing the expression of Ds. Conversely, the absence of Vg (and hence Fj) in neighboring pre-wing cells allows Ds expression and enhances its receiving capacity [76], creating an inductive Ft-to-Ds signaling interface (yellow/orange) with abutting wing cells that recruits the pre-wing cells into the wing. As depicted in the cell designated "Recruit," the Ft-to-Ds signal from the wing cell induces the unconventional myosin D to suppress the Wts kinase in the abutting pre-wing cell undergoing recruitment: This allows Yki to enter the nucleus and transduce the signal by activating the QE in response to Dpp and Wg (activating and inhibitory inputs are shown in solid black; absence of these inputs is depicted in outlines). In sum, as Dpp and Wg spread outwards from their sources along the A/P and D/V boundaries, they act jointly to fuel the growth and recruitment of pre-wing cells in front of the FF signaling interface, while at the same time sustaining the growth and specification of wing cells behind. Wing growth is thus governed by the progressive expansions in Dpp and Wg range. A/P, anterior–posterior; BE, Boundary enhancer; D/V, dorso–ventral; Dpp, Decapentaplegic; Ds, Dachsous; FF, feed-forward; Fj, Four-Jointed; Ft, Fat; PCP, planar cell polarity; QE, Quadrant enhancer; Sd, Scalloped; Vg, Vestigial; Wg, Wingless; Wts, Warts; Yki, Yorkie.

range, FF signal that acts together with Wg to induce neighboring pre-wing cells to grow, initiate QE-dependent *vg* transcription, and enter the wing (Fig 1E, "Recruit" cell). The newly recruited cells, like those already resident within the wing, then maintain QE-dependent Vg expression and grow in response to Wg. In addition, they become new sources of FF signal for the next round of *vg* induction. Accordingly, the outward spread of Wg from D/V border cells sustains the growth of wing cells behind the FF signaling interface while also inducing the growth and recruitment of pre-wing cells in front (Fig 1D).

**Experimental criteria for the FF paradigm for Wg.** In the absence of *ap*, the D/V segregation does not occur and the initial population of pre-wing cells ceases to express Vg, resulting in the *ap* (no wing) phenotype. Such *ap* null (*ap^o*) discs constitute a blank slate for dissecting the requirements for Wg and FF signaling, as follows.

First, the joint requirements for Wg and FF signaling in the recruitment and growth of QE-dependent wing tissue can be demonstrated by generating clones of cells within *ap^o* discs that express ectopic Wg, ectopic FF signal, or both. Regardless of the many diverse ways ectopic Wg and FF inputs can be supplied, pre-wing cells only initiate QE-dependent *vg* expression, convert to wing cells, and acquire the capacity to propagate the growth and recruitment of neighboring pre-wing tissue if they receive both [64–66]. For simplicity, we refer to the Wg-dependent induction and expansion of QE-dependent wing tissue as "FF growth."

Second, the dependence of FF growth on the extent of Wg spread can be demonstrated by supplying *ap^o* discs with seed populations of ectopic Vg-expressing cells that send FF signal and co-express either free Wg, a membrane tethered form of Wg (Neurotactin-Wg, Wg^Nrt [18]), or a constitutively active, intracellular transducer of Wg (myristoylated, truncated Armadillo, Arm* [18]). All such Vg-expressing cells grow cell-autonomously as wing tissue. However, cells that supply free Wg induce FF growth that can extend up to 30 or more cell diameters away, whereas cells that express Wg^Nrt induce FF growth that is restricted to neighboring cells and cells that express Arm* have no effect on the surround [64–66]. These results establish that the extent of FF growth depends on Wg range.

Third, the Vg-dependent, Vg-inducing FF signal has been identified as the protocadherin Ft, potentiated by the ectokinase Four-jointed (Fj [75]) and transduced by the conserved Hippo/Warts tumor suppressor pathway [88–92], specifically by the protocadherin Dachsous (Ds), the atypical myosin Dachs (D), the NDR kinase Warts (Wts) and the transcriptional coactivator Yorkie (Yki) (Fig 1E and 1F and legend) [66]. Manipulations of Ft, Fj, Ds, D, Wts, or Yki that generate ectopic Ft signal or constitutively activate the Ds/D/Wts/Yki transduction pathway are sufficient to substitute for ectopic Vg-expressing cells to initiate FF growth in *ap^o* discs provided that the responding cells are supplied with Wg [66]. Conversely, the capacity of wing cells to induce FF growth is greatly reduced when this signal transduction pathway is impaired [66].

**Testing the role of Dpp.**   Although *ap^o* discs fail to undergo D/V compartmentalization, they are subdivided into A and P compartments, with A cells along the compartment boundary expressing Dpp (Fig 2A). Hence, our previous studies of Wg-dependent FF growth were performed in the presence of Dpp, leaving the role and mode of action of Dpp untested. Here, we rectify this omission by applying the same experimental approaches used to establish the FF model for Wg to test the contribution of Dpp.

To do so, we use *ap^o* discs that are transheterozygous for 2 *dpp* enhancer deletion alleles, *dpp^d8* and *dpp^d10*, a genotype that precludes Dpp from being expressed in the wing disc [20,93], but does not compromise animal viability. The resulting, double mutant *dpp^d ap^o* discs thus constitute a more stringent "blank slate" that lacks not only a seed population of Vg-expressing Wg-secreting wing cells, but also a source of Dpp (Fig 2A). Accordingly, we can dissect the requirements for Dpp, Wg, and the FF signal by supplying them back in diverse combinations and ways as in our initial studies of Wg.

The results of the experiments described herein can be summarized simply. Dpp and Wg behave indistinguishably by every experimental criterion used to establish the FF hypothesis: Both are required, equally, to induce and sustain QE-dependent wing growth via their abilities to spread and to fuel the intra- and intercellular circuits of Vg autoregulation.

## Dpp is essential for seed populations of D/V border cells to induce FF growth

**Summary of approach and initial evidence for a requirement for Dpp.**   As in our previous studies [64–66], we manipulate Dpp, Wg, and/or FF signaling in *dpp^d ap^o* discs either (i) uniformly; (ii) in early induced, genetically marked clones; or (iii) in both ways. We then assay for FF growth, which we define as the nonautonomous induction and expansion of wing tissue specified by QE-dependent *vg* expression. In most experiments, we induce genetically marked clones that ectopically express the relevant factors by combining the Gal4/UAS [94,95] and Gal80 [96] techniques with the excision of transcriptional stop cassettes from "Flp-out" transgenes [97] (e.g., converting an inactive *UAS>stop>dpp* transgene to an active *UAS>dpp* transgene [10,17]; Materials and methods). When 2 or more Flp-out transgenes are present, any given disc can contain clones of different type depending on the combinations of resident Flp-out transgenes that have been activated by cassette excision. This allows us to assess the joint requirements for 2 or more signals on the nonautonomous induction of FF growth, which would otherwise not occur in *dpp^d ap^o* discs. We monitor FF growth by assaying for cells that express (i) either a *1XQE.lacZ* or *5XQE.DsRed* reporter transgene (both reliable indicators of QE-dependent *vg* expression); or (ii) native Vg protein, which is expressed in the Tsh^OFF Hth^OFF region of *ap^o* discs under the sole control of the QE [64–66]. We also assay for activation of the Dpp and Wg transduction pathways associated with FF growth by monitoring transcription factors that are expressed under the control of either Dpp (Spalt, Sal [17,98];

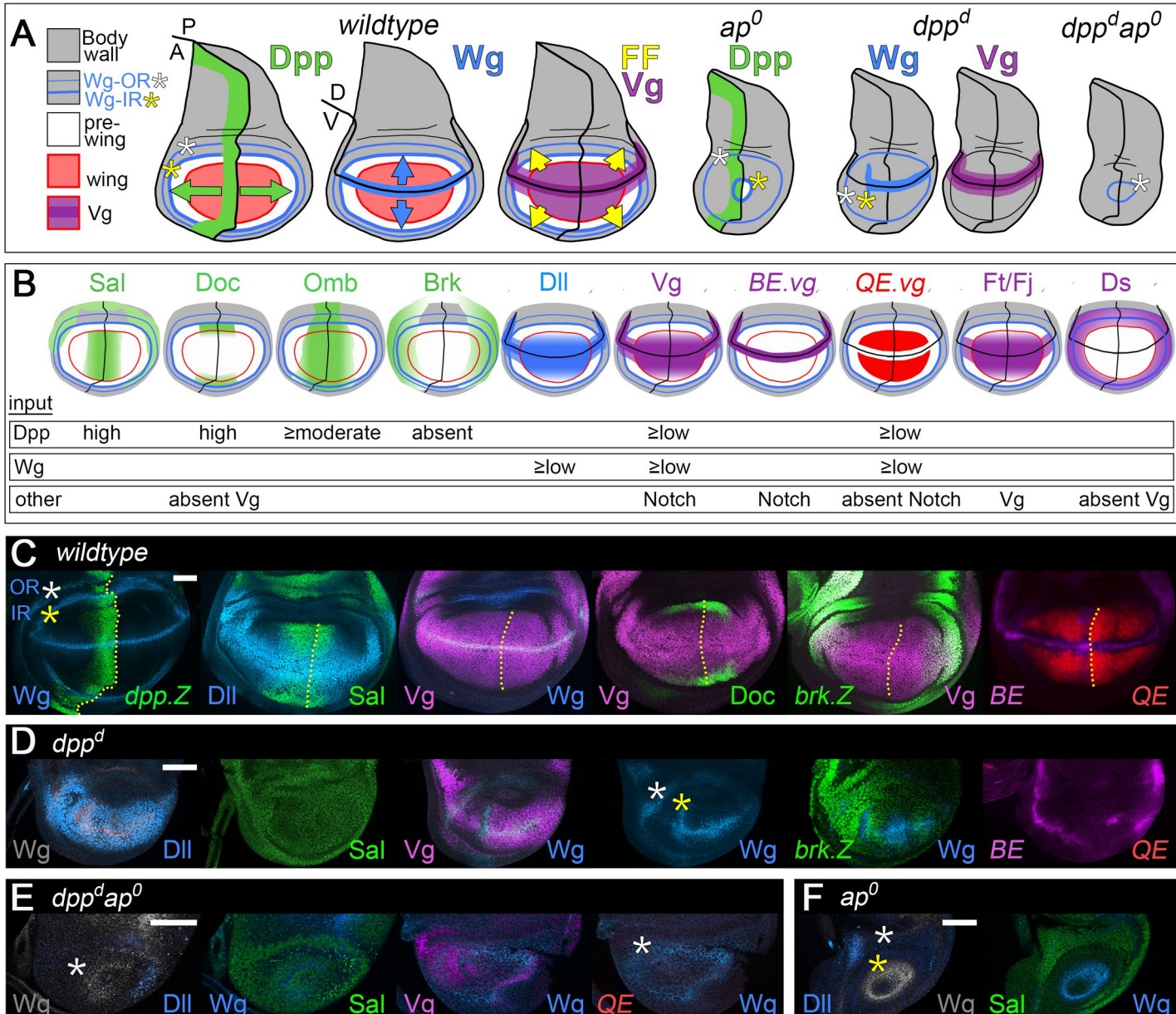

**Fig 2. Target gene expression in *wild-type*, *ap^o^*, *dpp^d^*, and *dpp^d^ ap^o^* discs.** (A) Sources and ranges of Dpp, Wg, and the FF signal in *wild-type*, *ap^o^*, *dpp^d^*, and *dpp^d^ ap^o^* wing discs in the mature third instar wing disc. The body wall domain, including the prospective hinge territory (gray) encircles the pre-wing (white) and wing (red) domains (the wing perimeter is outlined in darker red). Dpp, Wg, and the FF signal are color coded green, blue, and yellow, respectively, as in Fig 1. In *ap^o^* discs, no D/V segregation occurs, precluding Wg and Vg expression, production of the FF signal, and specification and growth of the wing, despite Dpp signaling from the A/P boundary (a small pre-wing domain, white, typically remains, encircled by "inner" and "outer" rings (IR and OR) of Wg expressing cells in the prospective hinge). In *dpp^d^* discs, Dpp is not produced along the A/P boundary, and the nascent pre-wing domain fails to develop despite the induction of Wg and Vg along the D/V boundary as well as expression of Wg-IR and Wg-OR cells in the hinge. Similarly, the pre-wing does not develop in *dpp^d^ ap^o^* discs, owing to the absence of Dpp, Wg, Vg, and the FF signal. The *dpp^d^ ap^o^* discs thus represent the null condition for determining the joint requirements for Dpp, Wg, and the FF signal. (B) The normal patterns of target genes used to assay Dpp, Wg, and FF signaling are depicted in green, blue, and red/purple, respectively, above a summary of the inputs required for their expression. (C, F) Documentation of the patterns of target gene expression in *wild-type* (C), *dpp^d^* (D), *dpp^d^ ap^o^* (E), and *ap^o^* (F) discs, color coded here and in subsequent figures as in B. Only the portions of the discs containing the pre-wing domain are shown, and the A/P boundary as well as the Wg-IR and Wg-OR are indicated by a dotted yellow line and yellow and white asterisks, respectively. Note that Sal is expressed at high level in response to Dpp in the wing and pre-wing territories, as well as at lower level, independent of Dpp, in the surrounding body wall. Note also that the patterns of Dll, Sal, and Wg in *dpp^d^* discs correspond to their normal expression in cells flanking the D/V boundary in the prospective hinge of *wild-type* discs, a context in which Wg produced by D/V border cells appears to act nonautonomously to extend the ranges of Dll up to several cell diameters away. Vg is also expressed together with Dll in the prospective hinge/body wall of *dpp^d^* as well as *dpp^d^ ap^o^* discs (and hence, outside of the purview of this study); this Vg expression is not mediated by either the QE or BE. Here, and in all subsequent figures, the scale bars indicate 50 microns. A/P, anterior–posterior; *ap*, *apterous*; BE, Boundary enhancer; D/V, dorso–ventral; Dpp, Decapentaplegic; FF, feed-forward; IR, inner ring; OR, outer ring; QE, Quadrant enhancer; Vg, Vestigial; Wg, Wingless.

Optomotor blind, Omb [17,99]; Dorsocross, Doc [100,101]; phosphorylated Mothers against Dpp, pMad [102]; Brinker, Brk [48,55,56,103]) or Wg (Distalless, Dll [18,19]). The expression patterns of all of these target genes as well as their requirements for Dpp, Wg, and other inputs are summarized in Fig 2.

As a prelude to assessing if seed populations of Wg-secreting, Vg-expressing D/V border cells require Dpp to induce FF growth, we generated clones of cells that express an N-terminally truncated, constitutively active form of Notch, N* [104], under Gal4/UAS control in *ap^o* discs that are *wild-type* for *dpp* (Materials and methods) and asked if the resulting FF growth is associated with endogenous Dpp spreading from the anterior–posterior (A/P) compartment boundary. As previously shown [64–66], such *UAS>N** clones behave as ectopic Wg and Vg-expressing D/V border cells and induce FF growth that is apparent as ectopic wing primordia marked by *5XQE.DsRed* expression [Fig 3A–3C; note that Notch also acts cell-autonomously to block QE-dependent *vg* expression (Fig 1E, Border cell; Fig 2B); hence, *UAS>N** clones can be identified by the absence of expression of QE-dependent reporter transgenes, as well as by the expression of Wg and native Vg (under BE control)]. Significantly, both Sal (indicative of high-level Dpp transduction [17,98]) as well as Dll (indicative of Wg transduction [13,18]) are expressed in the ectopic wing tissue, with the expression of both reporter proteins centered over the intersect between A compartment cells along the A/P boundary (the endogenous source of Dpp) and the *UAS>N** clones (the source of ectopic Wg; Fig 3B and 3C; here, and in subsequent figures, the sources of Dpp and Wg are represented in the accompanying diagrams by dark green and blue color, and their inferred ranges by regions of light green and light blue shading; key genotypes and reporter proteins are presented in the figure legends). These results provide circumstantial evidence that Dpp is required together with Wg to induce and sustain the growth of QE-dependent wing tissue.

**Testing the requirement for Dpp.** To determine if there is an obligate requirement for Dpp in addition to Wg, we tested the capacity of *UAS>N** clones to induce FF growth in *dpp^d ap^o* discs, which lack Dpp as well as D/V border cells. As expected, such clones ectopically express both Wg and BE-dependent Vg. However, they do not induce QE-dependent wing growth in abutting cells, as indicated by the absence of QE expression in the surround (*1xQE. lacZ* in Fig 3D: *5XQE.DsRed* in Fig 3E and 3F). As a positive control for this negative result, we repeated the experiment, but this time generating *UAS>N* UAS>dpp^GFP* clones, which co-express a biologically active, GFP-tagged form of Dpp in addition to N* (Materials and methods). Such *UAS>N* UAS>dpp^GFP* clones induce FF growth in *dpp^d ap^o* discs, as visualized by the expression of endogenous Vg as well as *5XQE.DsRed* (Fig 3G). These clones also induce both Sal and Dll in the surround (Fig 3H), indicating that their ability to induce FF growth correlates with their capacity to provide Dpp as well as Wg. Finally, we note that ectopic expression of Dpp is not sufficient to induce FF growth in *dpp^d ap^o* discs in the absence of *UAS>N** clones (corroborated in Fig 2A and 2F by the absence of FF growth in single mutant *ap^o* discs, which retain normal Dpp signaling). These results indicate that Dpp is required together with Wg for D/V border cells to induce and sustain FF growth.

## FF growth induced by ectopic D/V border cells depends on Dpp range

To determine if the extent of FF growth induced by ectopic D/V border cells depends on the range of Dpp, whether provided by these or other cells, we generated *UAS>N* UAS>dpp^GFP* clones in *dpp^d ap^o* discs under conditions in which Dpp^GFP is either free to move away from expressing cells or restricted by co-expression of a morphotrap (^mTrap)—a transmembrane protein bearing an extracellular nanobody directed against GFP that sequesters Dpp^GFP at the cell surface without interfering with its capacity to signal [105].

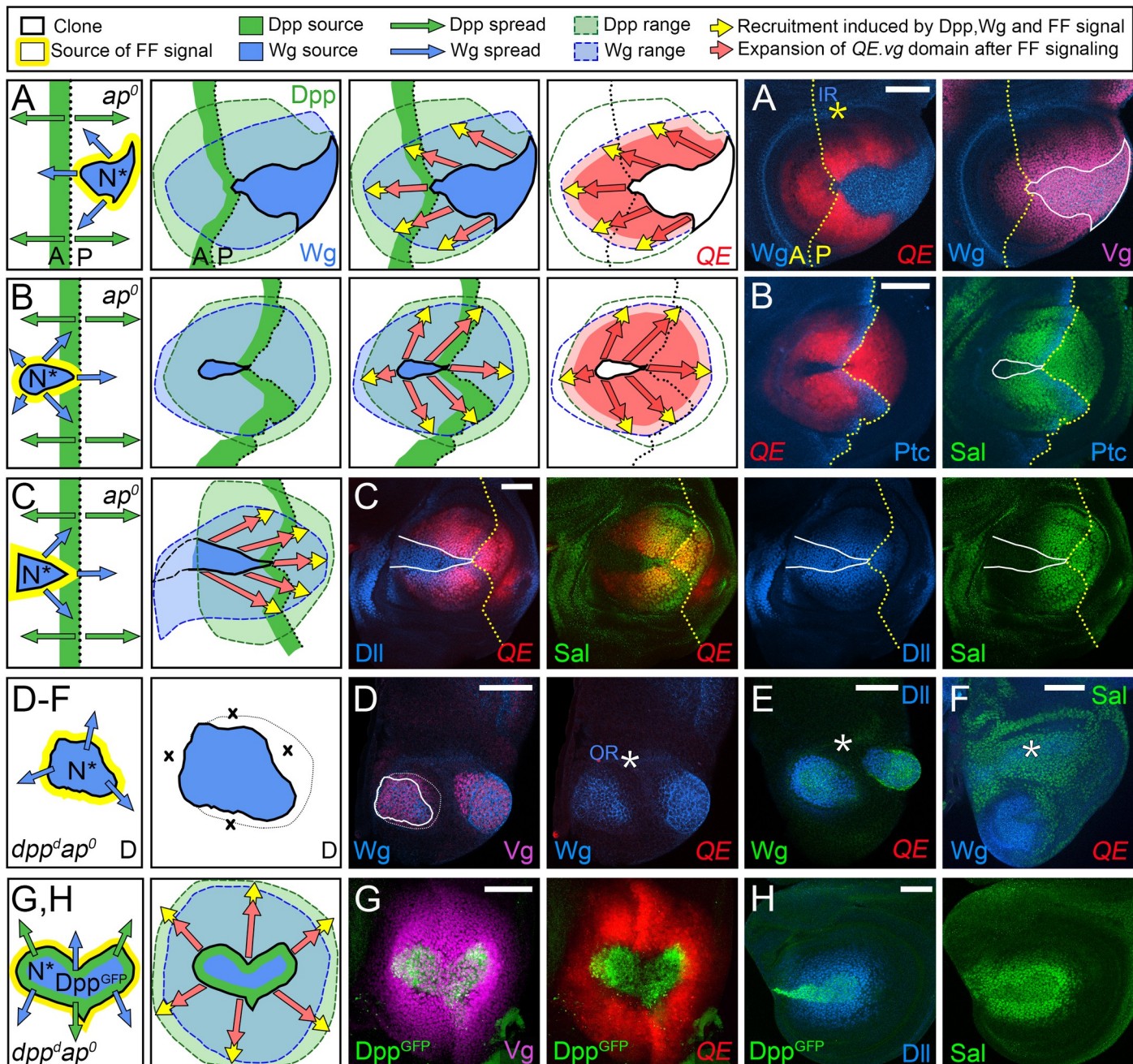

**Fig 3. Dpp is required for D/V border cells to induce FF growth. (A–C)** FF wing growth induced by ectopic D/V border cells in *ap⁰* discs correlates with the spread of Dpp (green) from A cells along the A/P boundary. As depicted in the left-most panels, clones of cells that express a constitutively active form of Notch under Gal4/UAS control (*UAS>N** clones) behave as seed populations of D/V border cells that express Wg (blue) and Vg, the latter generating the short-range Vg-dependent, Vg-inducing FF signal (yellow halo). For **A** and **B**, the cartoons in the middle panels depict the outputs of the clones. The left cartoons indicate the sources and inferred ranges of Dpp and Wg (light blue and green washes); the central cartoon indicates the 2 modes of growth: (i) expansion of the population of wing cells recruited by FF signaling (red arrows); and (ii) the growth and recruitment of pre-wing cells by the FF signal (yellow arrows); the right cartoon indicates the domain of QE-dependent Vg-expressing wing cells (red), as well as pre-wing cells in the process of being recruited (pink) relative to the inferred plumes of Wg and Dpp. Here, and in all subsequent figures, the yellow arrows depict FF recruitment in progress, and not the Vg-dependent, Vg inducing FF signal itself, which is generated wherever Vg-expressing wing cells confront non-Vg-expressing pre-wing cells. The micrographs to the right document the results. In **A**, we monitor Wg (which marks the *UAS>N** clone, blue) and both *5XQE.DsRed* expression (red) and native Vg (purple). The A/P border (which coincides with a local reduction in *5XQE. DsRed* expression, as in Fig 2C, far right panel) is indicated in yellow. Native Vg expression is induced cell-autonomously within the clone, under BE control, and nonautonomously in the surround, under QE control. The IR of Wg expression (yellow asterisk) marks cells in the surrounding hinge domain. In **B**, the clone is marked "black" by the absence of *5XQE.DsRed*, and A cells along the A/P boundary are visualized by Ptc expression, blue, which coincides with that of Dpp induced by Hh coming from the P compartment. The right panel shows the expression of Sal, which is induced by high levels of Dpp corroborating the inferred spread of

Dpp. **C** shows a third example of a *UAS>N\** clone, depicted and marked as in **B**. Here and in subsequent figures, we show only the input conditions and arrows depicting the 2 modes of FF growth (maintenance, red, and recruitment, yellow) relative to the inferred spreads of Dpp and Wg. In this case, we monitored both Sal and Dll (an indicator of Wg input, blue) to provide evidence that *5XQE.DsRed* expression is observed where cells are exposed to both Dpp and Wg. **(D, F)** *UAS>N\** clones generated in *dpp^d ap^o* discs express both Wg (blue in **D, F**; green in **E**) and Vg under BE control (purple in **D**). However, unlike such clones generated in *ap^o* discs, which retain *wild-type dpp* activity, they fail to activate the QE or induce FF growth in surrounding cells (here, and in other figures, such failures are marked by "x's" in the diagram), as indicated by the absence of *1XQE.lacZ* (**D**) and *5XQE.DsRed* (**E, F**) expressing cells (the white asterisk indicates the position of the wg OR). Instead, these clones behave like D/V border cells in the hinge domain of *wild-type* discs, which are associated with short-range nonautonomous induction of both Vg and Dll. They also fail to induce high levels of Sal expression (**F**), although lower level Sal expression is observed in the surrounding hinge and body wall domains as in *wild-type* discs. **(G,H)** *UAS>N\** clones that co-express a biologically active, GFP-tagged form of Dpp (Dpp^GFP, green) under *UAS* control induce QE-dependent Vg expression and FF growth in *dpp^d ap^o* discs as monitored by both *5XQE.DsRed* and native Vg (**G**). Such *UAS>N\* UAS>dpp^GFP* clones also induce surrounding cells to express both Sal and Dll, corroborating that the resulting FF growth is associated with the outward spread of Dpp and Wg from the clone (**H**). **Key genotypes** (here and in subsequent figures, see Materials and methods for full genotypes): *Tub>Gal80>Gal4 UAS>N\** in either *ap^o* discs (**A–C**), or *dpp^d ap^o* discs (**D, F**), or *dpp^d ap^o UAS>dpp^GFP* discs (**G, H**). A/P, anterior–posterior; *ap, apterous*; BE, Boundary enhancer; D/V, dorso–ventral; Dpp, Decapentaplegic; FF, feed-forward; FRT, >, Flp recombinase targets; IR, inner ring; N, Notch; OR, outer ring; QE, Quadrant enhancer; Vg, Vestigial; Wg, Wingless.

For the control condition (absence of the morphotrap), we co-induced *UAS>N\* UAS>dpp^GFP* clones with either *UAS>N\** or *UAS>dpp^GFP* clones (Fig 4A–4C, Materials and methods). As expected, all such *UAS>N\* UAS>dpp^GFP* clones induced extensive FF growth, and the same was also observed for *UAS>N\** clones co-induced with *UAS>N\* UAS>dpp^GFP* clones, provided that the surrounding, otherwise *wild-type* cells were in position to receive Dpp^GFP. For example, in Fig 4A and 4B, a *UAS>N\* UAS>dpp^GFP* clone abuts a *UAS>N\** clone and FF growth (marked by *5XQE.DsRed*) is apparent in cells surrounding both clones provided they are located within an approximately 15 to 20 cell diameter range of Dpp^GFP produced by the *UAS>N\* UAS>dpp^GFP* clone (light green; inferred in Fig 4A, and documented in Fig 4B by the expression of an *omb.lacZ* transgene, which is induced by moderate to high levels of Dpp [17]). In Fig 4C, a small *UAS>N\* UAS>dpp^GFP* clone abuts a larger *UAS>dpp^GFP* clone and has induced an extensive halo of FF growth in the surround. Importantly, *UAS>N\** clones fail to induce FF growth in abutting cells that are located outside of the range of Dpp^GFP expressing cells, providing an internal control confirming the requirement for Dpp (Fig 4A and 4B; regions marked by "x").

In contrast, in the experimental condition (Dpp^GFP movement restricted by co-expression of the morphotrap), *UAS>N\* UAS>dpp^GFP UAS>^mTrap* clones co-induced with *UAS>dpp^GFP UAS>^mTrap* clones induced FF growth, but the response was limited to cells that either express the morphotrapped Dpp^GFP or are in close proximity to cells that do. In the example shown (Fig 4D), a *dpp^d ap^o* disc contains 3 *UAS>N\* UAS>dpp^GFP UAS>^mTrap* clones as well as additional *UAS>dpp^GFP UAS>^mTrap* clones. Although there is extensive FF growth, all of the *5XQE.DsRed* expressing cells either co-express the *UAS>dpp^GFP UAS>^mTrap* transgenes or are located within several cell diameters of cells that do.

As a further test of the correlation between Dpp range and the extent of FF growth, we performed an equivalent version of the morphotrap experiment (Fig 4D), but used a constitutively active form of the Dpp co-receptor Thickveins (Tkv^QD) that acts cell-autonomously [17] instead of morphotrapped Dpp^GFP. In this case, co-induction of *UAS>N\** and *UAS>tkv^QD* clones can result in *UAS>N\** as well as *UAS>N\* UAS>tkv^QD* clones that induce FF growth, but only if these clones abut *UAS>tkv^QD* clones. Moreover, all such FF growth is limited strictly to cells within the *UAS>tkv^QD* clones. For example, in the *dpp^d ap^o* disc shown in Fig 4E, 2 adjacent *UAS>N\* UAS>tkv^QD* clones have induced extensive FF growth in the surround, but only in cells belonging to a third, abutting *UAS>tkv^QD* clone (all 3 clones are marked by high-level pMad accumulation).

We conclude that the extent of FF growth induced by D/V border cells depends on the range of Dpp, whether endogenous Dpp produced by cells along the A/P boundary of *ap^o*

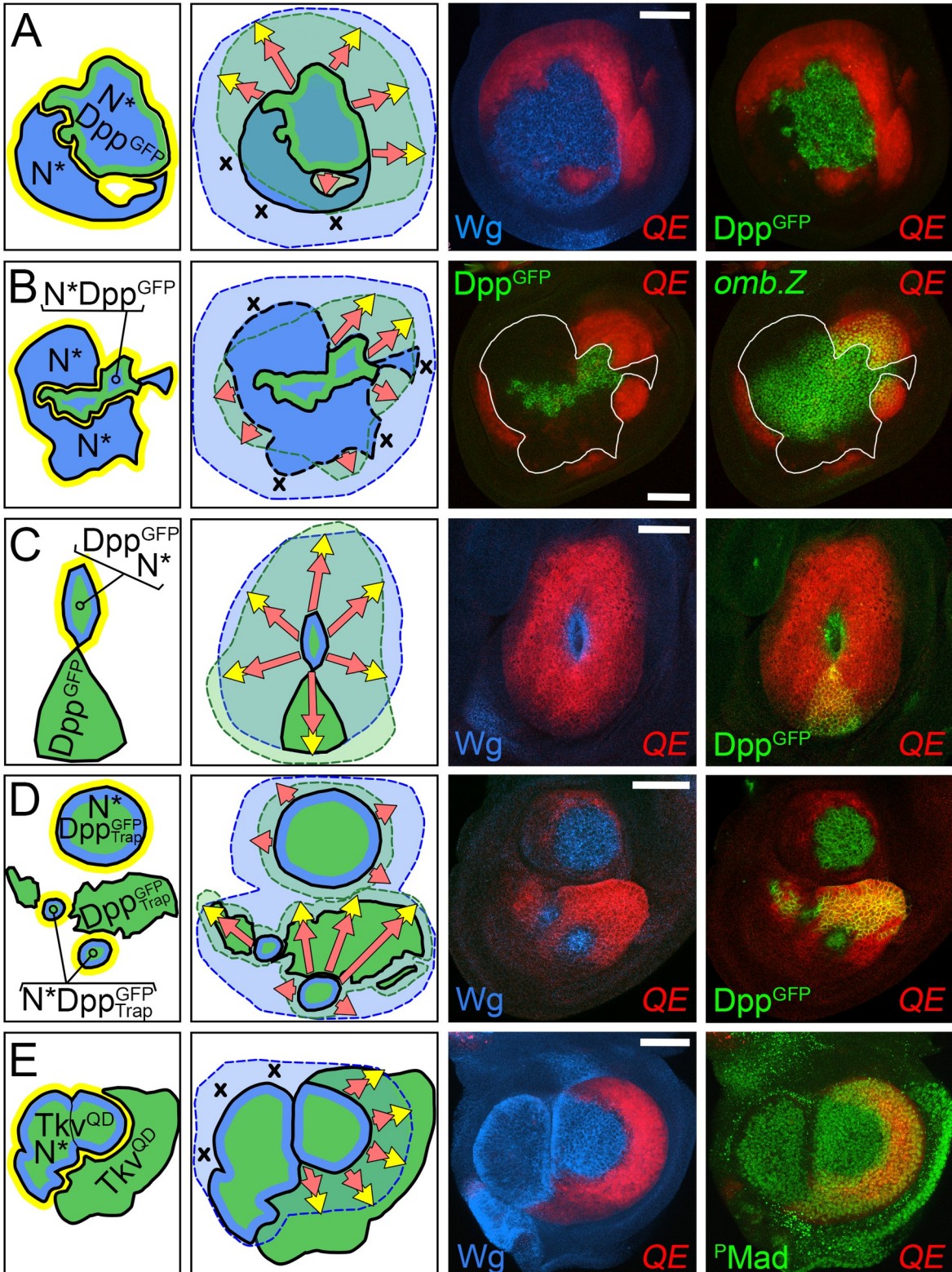

**Fig 4. FF growth induced by D/V border cells depends on Dpp range. (A–C)** FF growth induced in $dpp^d\ ap^o$ discs by the combined activities of $UAS>N^*$ and $UAS>N^*\ UAS>dpp^{GFP}$ clones is limited by the range of Dpp. In (**A**) and (**B**), the domain of FF growth (monitored by $5XQE.DsRed$ expression, red) is centered on a single $UAS>N^*\ UAS>dpp^{GFP}$ clone [marked by Dpp$^{GFP}$ (green)], irrespective of abutting $UAS>N^*$ clones (marked "black" by the absence of $5XQE.DsRed$, as well as by Wg (blue) in **A**]. In **B**, the range of

DppGFP is monitored independently by expression of an *omb.lacZ* transgene (green, right panel), which responds to moderate to high levels of Dpp input: *5XQE.DsRed* is expressed in cells that either express the *omb.lacZ* reporter or are located within several cell diameters of cells that do. However, cells that are located further away do not express *5XQE.DsRed* even when they abut clones of *UAS>N\** cells indicating that D/V border cells can only induce QE-dependent Vg expression and FF growth if the responding cells are in position to receive Dpp, as in **A**. **(C)** A *UAS>N\* UAS>dppGFP* clone that abuts a *UAS>dppGFP* clone has induced ectopic wing tissue (stained for *1XQE.lacZ* expression, red) that extends up to 30 cell diameters away, indicating the range over which Dpp can act to fuel FF growth (clones marked as in **A**; see also Fig 3G and 3H). **(D)** A *dppd apo* disc that carries multiple *UAS>dppGFP UAS>mTrap* clones that co-express DppGFP plus the morpho-trap (marked by DppGFP, green), some of which are also *UAS>N\** (marked by Wg, blue). The *UAS>N\** clones have induced extensive wing growth in the surround (monitored by *1XQE.lacZ* expression, red). However, this growth is limited to cells that are within or close to neighboring *UAS>dppGFP UAS>mTrap* clones, indicating that restricting the range of DppGFP by co-expressing the morphotrap reduces the extent of FF growth (compare with **A–C**). **(E)** A *dppd apo* disc that carries 3 *UAS>tkvQD* clones (marked by Phospho-Mad, green), two of which are also *UAS>N\** (marked by Wg. blue). The *UAS>N\** clones have induced FF growth (represented as *5XQE.DsRed* expression, red) that is strictly limited to cells in the abutting *UAS>tkvQD* clone. **Key genotypes: (A, B)** *dppd apo* discs that are also *Tub>Gal80>vg C765.Gal4 UAS>N\* UAS>stop>dppGFP* (note that this genotype generates 2 types of *UAS>N\** clones: those that do and those that do not co-express *UAS>dppGFP*, both of which express peak levels of endogenous Vg under BE control as well as low levels of exogenous Vg from excision of the *Tub>Gal80>vg* transgene: this low level exogenous Vg is gratuitous in the context of the peak native Vg expression). **(C)** *Tub>Gal80>Gal4 UAS>stop>N\* UAS>dppGFP* (this genotype generates 2 types of *UAS>dppGFP* clones: those that do and those that do not express *UAS>N\**). **(D)** *Tub>Gal80>Gal4 UAS>stop>N\* UAS>dppGFP UAS>mTrap* (as in **C**, except all clones also co-express *UAS>mTrap*). **(E)** *Tub>Gal80>Gal4 UAS>stop>N\* UAS>stop>tkvQD*; this genotype generates 3 types of clones: *UAS>N\**, *UAS>tkvQD*, and *UAS>N\* UAS>tkvQD*. *ap, apterous*; BE, Boundary enhancer; D/V, dorso–ventral; Dpp, Decapentaplegic; FF, feed-forward; FRT, >, Flp recombinase targets; mTrap, Morphotrap; N, Notch; QE, Quadrant enhancer; Tkv, Thickveins; Vg, Vestigial; Wg, Wingless.

discs (Fig 3A–3C), or from ectopic Dpp expressing cells in *dppd apo* discs, which are devoid of endogenous Dpp (Fig 4).

## FF growth induced by Vg-expressing cells depends on Dpp as well as Wg

Previously, we demonstrated that Notch activation initiates Wg-dependent FF growth via BE-dependent Vg activity [64–66]. To do so, we showed that clones that ectopically express Vg in *apo* discs bypass the requirement for Notch activation, provided (i) that they express sufficiently high levels of Vg to generate a robust FF signal; and (ii) that the abutting pre-wing cells are also supplied with Wg. We now ask (i) if the ability of such ectopic Vg-expressing cells to induce FF growth depends equally on Dpp as well as Wg; and (ii) if recruitment occurs via the up-regulation of QE-dependent Vg expression by Dpp acting in concert with Wg.

**Co-requirement for Dpp and Wg for FF growth induced by Vg-expressing cells.** To determine if Vg-expressing clones require Dpp as well as Wg to induce FF growth, we first generated clones of cells that express a biologically active, GFP-tagged form of Vg (VgGFP) under the direct control of the *Tubulinα1* (*Tub*) promoter in *dppd apo* discs supplied with uniform Dpp, WgNrt, or both (Materials and methods; Fig 5A–5C). All such *Tub>vgGFP* clones express levels of VgGFP similar to the peak levels of endogenous Vg (Fig 5A, legend). When generated in *dppd apo UAS.dpp UAS>wgNrt* discs, they induce extensive FF growth, as visualized in Fig 5A by monitoring endogenous Vg as well as the exclusion of Doc, the latter of which is normally expressed in pre-wing cells in response to high-level Dpp but repressed in wing cells by Vg ([101], Fig 2B and 2C; see also S1A and S1B Fig, which document that FF growth induced by such clones is restricted to the TshOFF HthOFF pre-wing territory of the wing disc). In contrast, *Tub>vgGFP* clones generated in *dppd apo* discs that ubiquitously express only *UAS>wgNrt* or *UAS.dpp* do not induce FF growth, as indicated by their failure to induce endogenous Vg expression in the surround and corroborated by their failure to exclude Doc expression from neighboring cells in *dppd apo UAS.dpp* discs (Fig 5B and 5C; Doc is not expressed in *dppd apo UAS> wgNrt* discs owing to the absence of Dpp).

As a further test of the joint requirement for Dpp and Wg, we repeated these experiments, only this time co-inducing *Tub>vgGFP* clones with *UAS>wgNrt* clones in the presence of uniform Dpp, or vice versa, co-inducing them with *UAS>tkvQD* clones in the presence of uniform

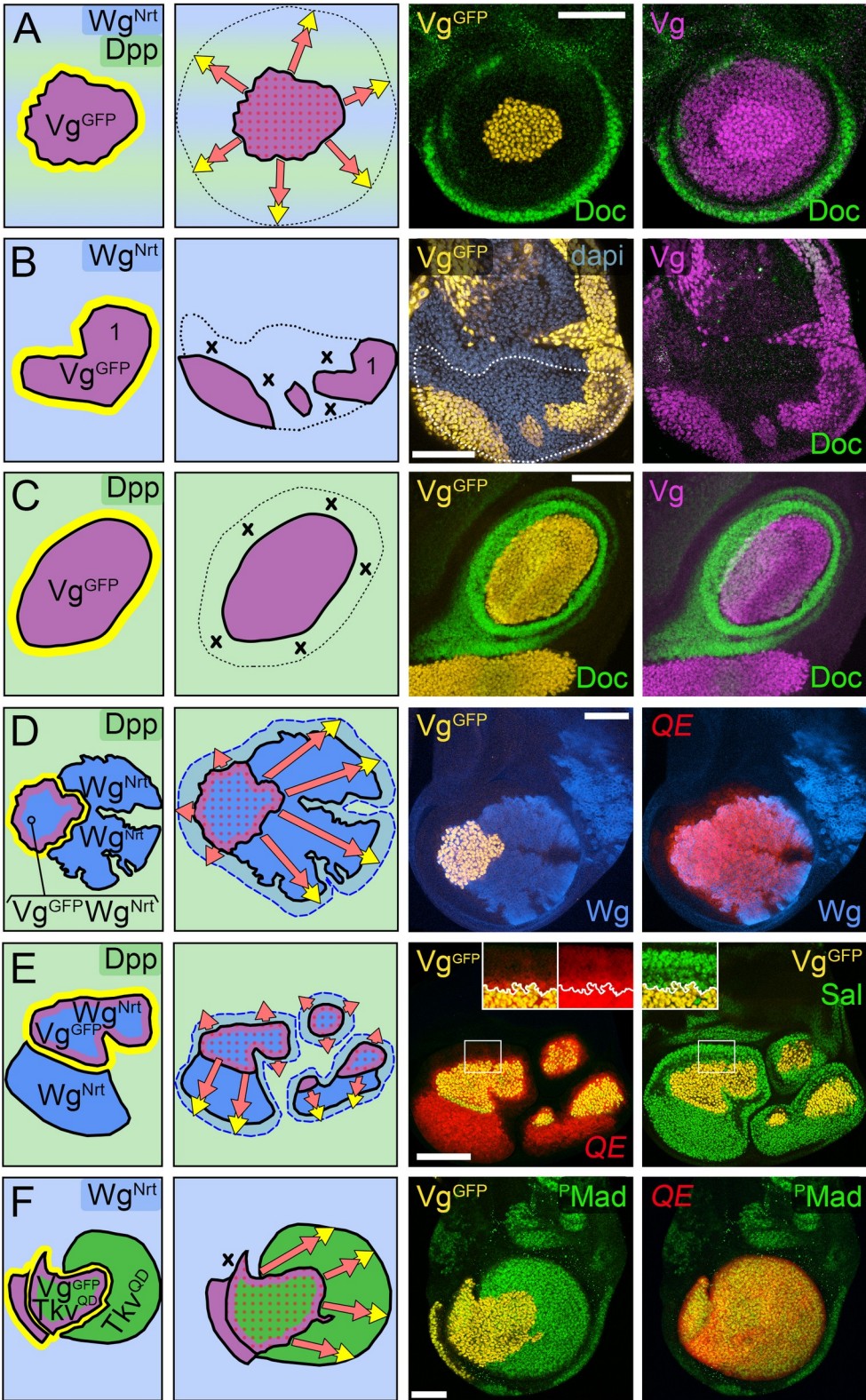

**Fig 5. Dpp is required together with Wg for the induction of FF growth by ectopic Vg-expressing cells. (A–C)** *Tub>vg^{GFP}* clones (marked by GFP, yellow) in *dpp^d ap^o* discs that uniformly express both Dpp and Wg^{Nrt} (**A**), only Wg^{Nrt} (**B**), or only Dpp (**C**) under Gal4/UAS control, stained for Vg (purple) and Doc (green); the dotted lines in the cartoons in **A–C** and the micrograph in **B** indicate a characteristic fold that delimits pre-wing from more proximal

hinge tissue (only 1 of 3 clones, #1, in **B** is depicted in the cartoon on the left). $Tub{>}vg^{GFP}$ clones generated in the presence of both Wg$^{Nrt}$ and Dpp induce FF growth, as indicated by the expression of native Vg and the exclusion of Doc in cells surrounding the clone in **A**. In contrast, clones generated in the presence of only Wg$^{Nrt}$ or Dpp fail to induce FF growth, as indicated by the absence of native Vg-expressing cells in the surround. Doc is absent in the $UAS{>}wg^{Nrt}$ disc (**B**), as expected, given that Doc expression is dependent on Dpp. Conversely, Doc is expressed in all cells surrounding the $Tub{>}vg^{GFP}$ clone in the *UAS.dpp* disc (**C**), corroborating both the uniform provision of Dpp, as well as failure of the clone to induce Vg expression in the abutting cells, which would otherwise repress Dpp-dependent Doc expression. Here and in Figs 5D–5F and 6–10 and S2 and S3 Figs, we depict Dpp plus Wg dependent QE activity that is located inside ectopic $Tub{>}vg$ or $Tub{>}vg^{GFP}$ clones, or in discs in which Yki is constitutively active ($ft^o$, $wts^o$ or *UAS.yki*) as a red dotted grid. In **A**, these territories are defined by the $Tub{>}vg^{GFP}$ clones and exhibit Vg staining that is approximately 2-fold higher than in the surround, indicating the joint and quantitatively similar contributions of the transgene and the 2 *wild-type* alleles of the endogenous gene. **(D, E)** $dpp^d$ $ap^o$ discs that express Dpp uniformly under Gal4 control and contain multiple clones that express Wg$^{Nrt}$ (blue), one, or a few of which also express Vg$^{GFP}$ under *Tub* control (yellow). The Vg$^{GFP}$ expressing clones induce extensive FF growth, as monitored by *5XQE.DsRed* (**D, E**) and Sal (**E**) expression in the surround, provided that the responding cells either express Wg$^{Nrt}$ or are positioned close to cells that do. Sal expression in **E** serves as an exceptionally sensitive proxy for Vg, as prospective wing cells that express barely detectable levels of the *5XQE.DsRed* reporter nevertheless express peak levels of Sal in response to Dpp (as documented in the inset by over-exposure of the *5XQE.DsRed* signal). **(F)** A $dpp^d$ $ap^o$ disc that expresses Wg$^{Nrt}$ uniformly and contains multiple $UAS{>}tkv^{QD}$ clones (marked by phospho-Mad, green), one of which also expresses Vg$^{GFP}$ (yellow). The Vg$^{GFP}$ expressing clone has induced extensive FF growth (monitored by *5XQE.DsRed*) that is strictly limited to cells that also express Tkv$^{QD}$. **Key genotypes:** $dpp^d$ $ap^o$ $Tub{>}stop{>}vg^{GFP}$*C765.Gal4* discs that are either *UAS.dpp* $UAS{>}wg^{Nrt}$ (**A**), or $UAS{>}wg^{Nrt}$ (**B**), or *UAS.dpp* (**C**), or that are *UAS.dpp* $UAS{>}stop{>}wg^{Nrt}$ (**D, E**) or $UAS{>}wg^{Nrt}$ $UAS{>}stop{>}tkv^{QD}$ (**F**). *ap*, *apterous*; Dpp, Decapentaplegic; FF, feed-forward; FRT, >, Flp recombinase targets; QE, Quadrant enhancer; Tkv, Thickveins; *Tub*, *Tubulinα1*; Vg, Vestigial; Wg, Wingless; Wg$^{Nrt}$, Neurotactin-Wingless; Wts, Warts; Yki, Yorkie.

Wg$^{Nrt}$. In both cases, we find that $Tub{>}vg^{GFP}$ clones can induce FF growth but only when the responding cells receive both Wg$^{Nrt}$ and Dpp/Tkv input. Fig 5D and 5E document this for $Tub{>}vg^{GFP}$ clones that are co-induced with $UAS{>}wg^{Nrt}$ clones in the presence of uniform Dpp. In both examples, $Tub{>}vg^{GFP}$ clones that co-express $UAS{>}wg^{Nrt}$ have induced QE-dependent Vg expression in neighboring pre-wing cells, and when adjacent to clones that express only $UAS{>}wg^{Nrt}$, they have induced FF growth that extends 30 or more cell diameters into the abutting clones as well as to neighboring, otherwise *wild-type* cells outside of the clone (as monitored by *5XQE.DsRed* expression in Fig 5D and 5E, and by Sal expression, an especially sensitive indicator of endogenous Vg activity in discs that uniformly express Dpp, in Fig 5E; see also S1C Fig). Similarly, $Tub{>}vg^{GFP}$ clones that are co-induced with $UAS{>}tkv^{QD}$ clones in the presence of uniform Wg$^{Nrt}$ can induce FF growth. However, in contrast to $Tub{>}vg^{GFP}$ clones co-induced with $UAS{>}wg^{Nrt}$ clones in the reciprocal experiment (Fig 5D and 5E), they do so only when they abut neighboring $UAS{>}tkv^{QD}$ clones; moreover, all such FF growth is restricted to cells within the $UAS{>}tkv^{QD}$ clone (Fig 5F). These differences are expected given the short-range, nonautonomous action of Wg$^{Nrt}$ [18,64] versus the strictly cell-autonomous action of Tkv$^{QD}$ [17].

These results establish that pre-wing cells must receive both Dpp input (whether supplied by Dpp itself or constitutive activation of Tkv) as well as Wg to initiate QE-dependent Vg expression in response to ectopic Vg-expressing cells.

**Requirements for Dpp and Wg in the up-regulation of Vg required to recruit pre-wing cells into the wing.** As summarized in Fig 1D and 1E, recruitment of pre-wing cells into the wing requires 2 distinct steps, the first being the induction of Yki activity in pre-wing cells to initiate QE-dependent Vg expression and the second being engagement of a positive circuit of Vg autoregulation that allows Vg to substitute for Yki and amplify its own expression to select the wing state. The preceding experiments assess both the sufficiency of Vg to generate a productive FF signal, as well as the joint requirements for Dpp and Wg to initiate QE-dependent Vg expression. Here, we address the requirements for Dpp and Wg to fuel the subsequent up-regulation of *vg* upon which recruitment depends. To do so, we used a $Tub{>}stop{>}vg$

transgene that produces barely detectable levels of untagged Vg protein following excision of the $>stop>$ cassette. When generated in $dpp^d ap^o$ discs, such $Tub>vg$ clones generate exogenous Vg, bypassing the normal requirements for Dpp, Wg, and the FF signal to initiate QE-dependent expression of endogenous Vg. However, they do not bypass the subsequent requirement for Vg to up-regulate its own expression via the endogenous *vg* gene and hence fail to adopt the wing state. To determine if Dpp and Wg are required for this autoregulatory amplification to occur, we co-induced such clones together with $UAS>dpp^{GFP}$ and $UAS>wg^{Nrt}$ clones—in this case focusing on the response of cells inside rather than outside the clones.

$Tub>vg$ clones that are generated on their own or co-induced with $UAS>dpp^{GFP}$ clones in $dpp^d ap^o$ discs express low levels of Vg derived predominantly if not exclusively from the transgene and show little if any detectable expression of the *5XQE.DsRed* reporter (corroborated in Fig 6A and 6B). Likewise, $Tub>vg$ clones co-induced with $UAS>wg^{Nrt}$ clones express similarly low levels of Vg and are devoid of detectable *5XQE.DsRed* expression (corroborated in Fig 6C and 6D). In contrast, when co-induced with both $UAS>dpp^{GFP}$ and $UAS>wg^{Nrt}$ clones, $Tub>vg$ cells that co-express both Dpp$^{GFP}$ and Wg$^{Nrt}$ express peak levels of Vg (Fig 6A) as well as the *5XQE.DsRed* reporter (Fig 6B and 6C), indicating peak, QE-dependent up-regulation of the endogenous *vg* gene. In addition, such high level, Vg-expressing cells can induce abutting, otherwise *wild-type* cells to express peak levels of both Vg and the *5XQE.DsRed*, provided the responding cells are also in position to receive both Dpp$^{GFP}$ and Wg$^{Nrt}$ (Fig 6A–6C).

For example, in Fig 6A, a large $Tub>vg$ $UAS>dpp^{GFP}$ clone expresses barely detectable levels of Vg, in contrast to a neighboring $Tub>vg$ $UAS>dpp^{GFP}$ $UAS>wg^{Nrt}$ clone that cell-autonomously expresses high levels of Vg and has induced abutting cells around its entire circumference to do the same. Essentially, the same result is documented in Fig 6B and 6C, except that *5XQE.DsRed* rather than Vg expression is assayed, and in Fig 6C, there are 2 $Tub>vg$ $UAS>wg^{Nrt}$ clones abutting the $Tub>vg$ $UAS>dpp^{GFP}$ $UAS>wg^{Nrt}$ clone. In this case, *5XQE.DsRed* expression has propagated through the 2 $Tub>vg$ $UAS>wg^{Nrt}$ clones as well as to neighboring, otherwise *wild-type* cells. However, cells within the right most $Tub>vg$ $UAS>wg^{Nrt}$ clone that are located farthest from the abutting $Tub>vg$ $UAS>dpp^{GFP}$ $UAS>wg^{Nrt}$ clone (marked by a white "x") do not express the *5XQE.DsRed* reporter, a failure we interpret as due to their being outside of the range of Dpp$^{GFP}$. This inference is corroborated in Fig 6D, which shows a $Tub>vg$ $UAS>dpp^{GFP}$ clone that has induced high-level endogenous Vg expression and extensive wing growth in most cells of an abutting $Tub>vg$ $UAS>wg^{Nrt}$ clone, except for cells that are located just beyond the range of Dpp$^{GFP}$ (marked by a white "x" and confirmed by the expression of a *brk.lacZ* [48] reporter in $UAS>wg^{Nrt}$ expressing cells, identifying these as cells that belong to the clone but have received too little Dpp$^{GFP}$ to repress the reporter).

Importantly, even though both Dpp and Wg are essential to drive the cell autonomous *vg* autoregulatory circuit, they are not sufficient to initiate it in the absence of FF signal. In particular, $dpp^d ap^o$ *UAS.dpp* $UAS>wg^{Nrt}$ discs lack a source of FF signaling and fail to express Vg under QE control despite having uniformly high levels of Dpp and Wg input—unless supplied with an ectopic source of FF signal (Fig 5A). Conversely, $dpp^d ap^o$ discs supplied with only uniform Dpp or uniform Wg$^{Nrt}$ fail to do so, even when presented with the same ectopic source (Fig 5B and 5C). Hence, both the initial induction as well as the subsequent up-regulation of QE-dependent Vg expression depend on Dpp plus Wg.

We conclude (i) that Vg activity is sufficient to program wing cells to send FF signal; (ii) that pre-wing cells must receive both Dpp and Wg to initiate and then up-regulate Vg expression in response to this signal; and (iii) that the extent of FF growth induced by Vg-expressing cells depends on the ranges of both Dpp and Wg.

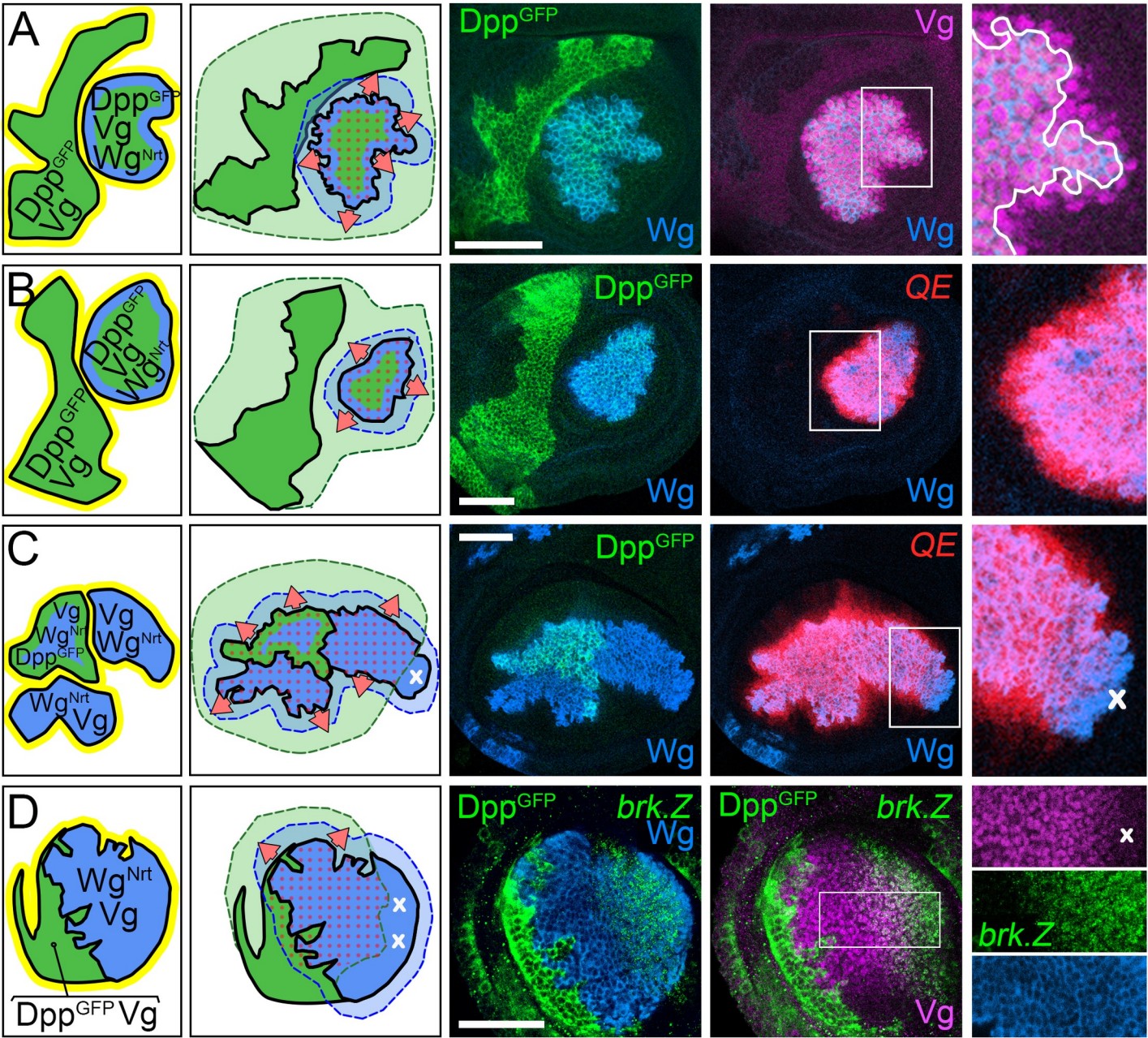

**Fig 6. Vg autoregulation required to recruit pre-wing cells into the wing depends on both Dpp and Wg. (A, B)** $dpp^d ap^o$ discs that carry 2 *Tub>vg* clones, one expressing *UAS>dpp^{GFP}* (green) and the other *UAS>dpp^{GFP}* plus *UAS>wg^{Nrt}* (blue), stained for Vg (**A**, purple) or *5XQE.DsRed* (**B**, red). Unlike *Tub>vg^{GFP}* clones, which constitutively express levels of Vg^{GFP} that correspond to peak levels of endogenous Vg (e.g., Fig 5A), cells within *Tub>vg UAS>dpp^{GFP}* clones express barely detectable levels of untagged exogenous Vg (**A**) unless they co-express *UAS>wg^{Nrt}* or abut clones that do, in which case they express peak levels due to positive autoregulation of the endogenous *vg* gene, and induce neighboring, *wild-type* cells to do the same (**A**, inset, the clone is outlined in white). The same result is obtained when such pairs of clones are monitored for *5XQE.DsRed* activity, except that the adjacent *Tub>vg* clone that co-expresses only *UAS>dpp^{GFP}* is devoid of detectable *5XQE.DsRed* activity (**B**). (**C**) A $dpp^d ap^o$ disc that carries 3 *Tub>vg* clones that express *UAS>wg^{Nrt}*, one of which also co-expresses *UAS>dpp^{GFP}*. All of the cells in these clones, as well as their abutting, otherwise *wild-type* neighbors, express peak levels of *5XQE.DsRed*, with the notable exception of cells in the right-most clone that are positioned furthest from the sole *UAS>dpp^{GFP}* clone (white x in diagram, documented in inset). (**D**) A $dpp^d ap^o$ disc that carries abutting *Tub>vg* clones, one of which expresses *UAS>dpp^{GFP}* while the other expresses *UAS>wg^{Nrt}*, monitored for expression of Vg (purple) as well as a *brk.lacZ* reporter (cytosolic green in contrast to membrane-associated green for Dpp^{GFP}) that is strongly repressed by Dpp. Most cells in, and next to, the *UAS>wg^{Nrt}* clone express peak levels of Vg, reflecting positive autoregulation of the native gene. However, the level of Vg expression fades to undetectable levels in cells furthest from the *UAS>dpp^{GFP}* clone, complementary to a rise in *brk.lacZ* expression (inset), which is inversely proportional to Dpp input. Note also the dramatic expansion of the population of *Tub>vg UAS>wg^{Nrt}* wing cells relative to that of the population of the abutting *Tub>vg UAS>dpp^{GFP}* wing cells, illustrating the selective induction of FF wing growth in cells that receive both Wg and Dpp, as opposed to cells that receive just Dpp or just Wg. Taken together, these results indicate an absolute requirement for both Dpp and Wg in the QE-

dependent autoregulation of endogenous *vg* primed by ectopic, low level expression of the *Tub>vg* transgene. **Key genotypes:** *dpp^d ap^o Tub>Gal80>vg C765.Gal4* discs that are also *UAS>dpp^{GFP} UAS>stop>wg^{Nrt}* (**A**, **B**), or *UAS>stop>dpp^{GFP} UAS>wg^{Nrt}* (**C**), or *UAS>stop>dpp^{GFP} UAS>stop>wg^{Nrt}* (**D**). *ap*, apterous; Dpp, Decapentaplegic; FF, feed-forward; FRT, >, Flp recombinase targets; QE, Quadrant enhancer; Vg, Vestigial; Wg, Wingless; Wg^{Nrt}, Neurotactin-Wingless;.

## FF growth induced by Fat/Dachsous signaling depends on Dpp as well as Wg

In the FF paradigm as elucidated for Wg, the protocadherins Ft and Ds function as a ligand/receptor pair, with the "on" and "off" states of Vg in wing versus pre-wing cells creating a Ft/Ds signaling interface between the 2 cell populations [66]. This is achieved via the capacity of Vg to drive the expression of Fj, an ectokinase that cell-autonomously potentiates the signaling activity of Ft while suppressing the responding activity of Ds [75,76] and by the ability of Vg to block Ds expression (Fig 1F, legend). The result is a confrontation between Fj expressing wing cells programmed to send Ft signal and Ds-expressing pre-wing cells programmed to receive it, inducing the latter to activate QE-dependent Vg expression—provided that they also receive Wg (as in Fig 1E and 1F; [66]).

To determine if pre-wing cells must also receive Dpp as well as Wg to respond to Ft, we have performed 2 experiments. In the first, we induced *UAS.ft* clones that co-express either *UAS>dpp^{GFP}*, *UAS>wg^{Nrt}* or both in *dpp^d ap^o* discs, and asked if the elevated level of Ft expression within the clone can substitute for the absence of Fj and allow Ft to induce Vg expression in abutting pre-wing cells, provided that the responding cells also receive both Dpp and Wg. Note that prior to the induction of such *UAS.ft* clones, all of the cells are in the pre-wing state, and hence express high levels of Ds and little or no Fj. Accordingly, they are primed to respond to Ft from cells within such clones.

As expected, *UAS.ft* clones that co-express only *UAS>dpp^{GFP}* or *UAS>wg^{Nrt}* fail to induce Vg expression (corroborated in regions that abut individual *UAS.ft UAS>dpp^{GFP}* or *UAS.ft UAS>wg^{Nrt}* clones, depicted by "x's" in the cartoons in Fig 7A and 7B). In contrast, *UAS.ft UAS>dpp^{GFP} UAS>wg^{Nrt}* clones can induce abutting cells to express Vg (Fig 7A and 7B). *UAS.ft UAS>wg^{Nrt}* clones can also induce abutting, otherwise *wild-type* cells to express Vg, provided that the latter are located within 15 to 20 cell diameters of independent clones that express *UAS>dpp^{GFP}* (Fig 7B).

The second experiment is similar to the first, except we used *UAS.wg* instead of *UAS>wg^{Nrt}* to generate free Wg (Fig 7C) rather than membrane tethered Wg^{Nrt} (Fig 7A and 7B). Again, we see that *UAS.ft* clones only induce Vg expression in abutting cells if they co-express both *UAS.wg* and *UAS>dpp^{GFP}* or abut cells that are in position to receive both Wg and Dpp^{GFP} from other clones. However, unlike the first experiment, Vg expression induced by such clones can extend many cell diameters into the surround (Fig 7C, inset), in contrast to being limited to cells in close proximity when such clones ectopically express Wg^{Nrt} rather than Wg (Fig 7A and 7B, insets).

Taken together, these experiments indicate that ectopic Ft signal is sufficient to induce Vg expression in abutting pre-wing cells, provided that the responding cells receive both Dpp as well as Wg.

Although the primary focus of these experiments is to test the joint requirements for Ft, Dpp, and Wg in inducing Vg expression in neighboring cells, it is notable that in most cases, *UAS.ft* clones that induce Vg in the surround also show Vg induction within (as in Fig 7A–7C). We envisage that this is due to a response of the *UAS.ft* cells either to a "return" FF signal coming from the abutting, pre-wing cells once they are induced to express Vg (depicted as yellow arrows in the cartoons to the right) or to initially heterogenous expression of ectopic Ft within the *UAS.ft* clones that creates transient local signaling interfaces. In support, no such

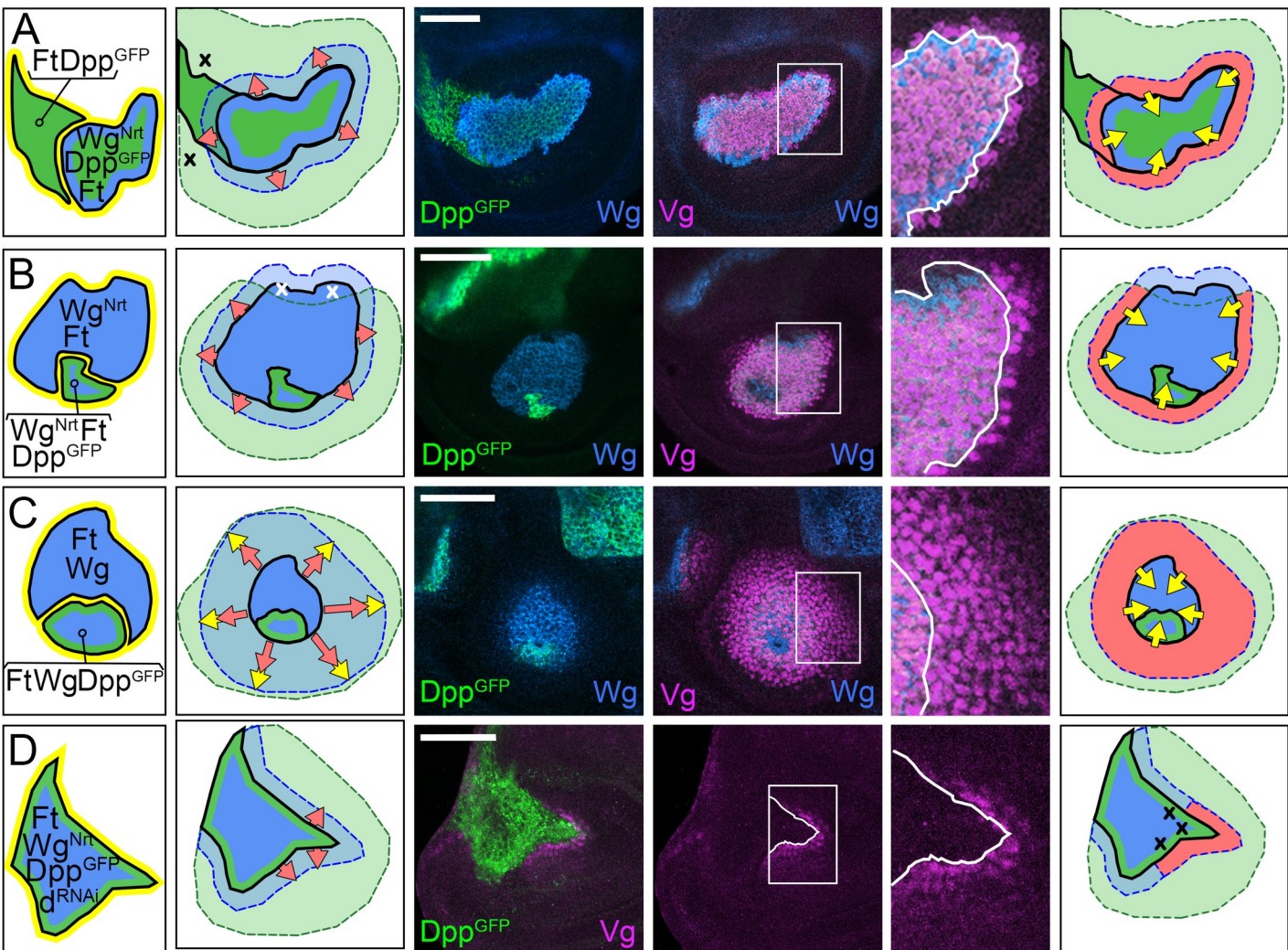

**Fig 7. Dpp is required together with Wg for the induction of FF growth by ectopic Ft expressing cells. (A, B)** *UAS.ft* clones that co-express either *UAS>dpp^{GFP}* (green), *UAS>wg^{Nrt}* (blue), or both in *dpp^d ap^o* discs. All such clones act nonautonomously to induce Vg expression (purple) in abutting cells, provided these cells are adjacent to *UAS>wg^{Nrt}* cells and within range of *UAS>dpp^{GFP}* cells. They also express Vg in most or all cells inside the clone, an unexpected finding that can be attributed to either a return Ft signal from surrounding *wild-type* cells (yellow arrows in the cartoons on the right), or to signaling within the clone caused by initially heterogenous expression of Ft within the clone (see text), as corroborated by the experiment in **D**. In **A**, the *UAS.ft UAS>dpp^{GFP}* clone to the left fails to induce Vg expression in otherwise *wild-type* cells in the surround, in contrast to the abutting *UAS.ft UAS>dpp^{GFP} UAS>wg^{Nrt}* clone to the right (inset; here and in the remaining insets, the clone border is marked in white). In **B**, a large *UAS.ft UAS>wg^{Nrt}* clone abuts a small *UAS.ft UAS>dpp^{GFP}UAS>wg^{Nrt}* clone and has induced Vg expression in otherwise *wild-type* cells around its entire circumference, with the notable exception of a small subset of cells that are located farthest from the *UAS.ft UAS>dpp^{GFP}UAS>wg^{Nrt}* clone (inset, top of the image). **(C)** A *dpp^d ap^o* disc that carries 2 *UAS.ft* clones, one co-expressing *UAS.wg* and the other co-expressing both *UAS.wg* plus *UAS>dpp^{GFP}* (stained as in **A**). In this case, ectopic Wg generated by the clones is not tethered, resulting in FF wing growth that extends many cell diameters into the surround in contrast to the local induction of Vg expression in abutting cells observed in **A, B** (insets). This FF growth is also dependent on Dpp^{GFP}, as peak Vg expression was only induced in and around *UAS.ft UAS.wg* clones if they either co-expressed *UAS>dpp^{GFP}* or were located within reach of a *UAS.ft UAS.wg* clone that did. **(D)** A *dpp^d ap^o* disc containing a single *UAS.ft UAS>dpp^{GFP}UAS>wg^{Nrt}* clone that co-expresses a *UAS.d^{RNAi}* transgene to knock down the ability of cells within the clone to respond to Ft signal (as in Fig 8). The clone has induced Vg in the surround, but unlike the *UAS.ft* clones shown in **A–C**, fails to show Vg expression within the clone. Note that in these experiments, we assay QE-dependent Vg expression by monitoring Vg protein itself, rather than *5XQE.DsRed* reporter expression. We do so, because we have previously shown that *UAS.ft* clones can induce low level *5XQE.DsRed* expression in abutting cells in *ap^o* discs that are *wild-type* for *dpp* [66]. We have previously attributed this response to the presence of a cryptic supply of Wg, which is sufficient in combination with native Dpp plus Ft signaling to weakly activate the QE but is not adequate to fuel the autoregulation of the *vg* locus on which the detection of Vg protein and specification of the wing state depends [*ibid*]. **Key genotypes:** *dpp^d ap^o Tub>Gal80>Gal4 UAS.ft* discs that are also *UAS>stop>dpp^{GFP} UAS>stop>wg^{Nrt}* (**A, B**), or *UAS>stop>dpp^{GFP} UAS.wg* (**C**), or *UAS>stop>dpp^{GFP} UAS>wg^{Nrt} UAS.d^{RNAi}* (**D**). ap, apterous; Dpp, Decapentaplegic; FF, feed-forward; FRT, >, Flp recombinase targets; Ft, Fat; QE, Quadrant enhancer; Vg, Vestigial; Wg, Wingless; Wg^{Nrt}, Neurotactin-Wingless.

Vg expression is observed when we selectively block the capacity of cells within *UAS.ft UAS>wg^{Nrt} UAS>dpp^{GFP}* clones to receive incoming Ft signal (Fig 7D), as described in the next section.

## FF growth fueled by Dpp and Wg depends on Ft signal transduction by Dachs

In the context of Wg-dependent FF growth, Ft signal is transduced by Ds via the atypical myosin D [66]. D acts downstream of Ds by preventing or reversing a conformational transition required for Wts kinase activity [106], allowing Yki, the transcriptional effector of the FF signal [88–90,92], to escape phosphorylation, enter the nucleus and activate the QE (Fig 1F). Hence, removing D results in constitutive phosphorylation of Yki by Wts, preventing Yki from activating the QE in response to Ft. Accordingly, by genetically ablating D, we can assess whether Dpp-dependent FF growth depends on transduction of the FF signal by Yki.

In a first series of experiments, we asked if *Tub>vg^{GFP}* clones can induce FF growth in *dpp^d d^o ap^o UAS>dpp^{GFP} UAS>wg^{Nrt}* discs, in which all cells are exposed to Dpp^{GFP} and Wg^{Nrt}, but devoid of D. Normally, *Tub>vg^{GFP}* clones induce FF growth in *dpp^d ap^o UAS.dpp UAS>wg^{Nrt}* discs; moreover, their ability to do so depends, in an absolute fashion, on the presence of both the *UAS.dpp* and *UAS>wg^{Nrt}* transgenes (Fig 5A–5C; corroborated for *UAS>dpp^{GFP}* rather than *UAS.dpp* in Fig 8A). In contrast, in the absence of D, *Tub>vg^{GFP}* clones autonomously express peak levels of QE-dependent Vg, but are deficient in their ability to induce QE activity and FF growth in the surround (Fig 8B). However, as previously observed, the effect is not absolute [66]. Instead, faint expression of QE-dependent Vg activity is detectable in *d^o* cells that abut the *Tub>vg^{GFP}* clones (indicated by low level expression of the *5XQE.DsRed* reporter and local exclusion of Doc in Fig 8B)—possibly due to an alternative mechanism by which Ft/Ds signaling can still reduce Wts activity, albeit only modestly, in the absence of D [107].

To distinguish whether D is required selectively in either wing or pre-wing cells, we performed a second set of experiments similar to the first, only this time removing D in clones of cells rather than in the entire disc. As shown in Fig 8C–8E, we find that *d^o* clones in *dpp^d ap^o UAS.dpp UAS>wg^{Nrt}* discs (marked "black" by the absence of GFP expression, yellow, and outlined in white) are severely limited in their ability to initiate QE-dependent Vg expression in response to *Tub>vg^{GFP}* clones (marked by bright yellow and outlined in yellow). Instead, they show only barely detectable, local induction of *5XQE.DsRed* (Fig 8C, red) and native Vg (Fig 8D and 8E, purple), as we observed in entirely *d^o* discs (Fig 8B). In contrast, *Tub>vg^{GFP}* clones that are *d^o* appear fully competent to induce FF growth of the surround. For example, the *d^o Tub>vg^{GFP}* clone shown in Fig 8F (outlined in white and yellow) has induced a surrounding halo of cells that expresses native Vg (dull purple) and extends up to 10 to 15 cell diameters away, whereas a small, abutting *d^o* clone (outlined only in white) has failed to respond (Fig 8F, see legend).

Finally, we have used the selective requirement for D in transducing Ft signal in pre-wing cells to test whether the unexpected activation of QE-dependent Vg expression observed inside *UAS.ft UAS>dpp^{GFP} UAS>wg^{Nrt}* clones (Fig 7A–7C) depends on Ft input, as proposed in the preceding section. Specifically, we generated *UAS.ft UAS>dpp^{GFP} UAS>wg^{Nrt}* clones that co-express a *UAS.d^{RNAi}* transgene that severely reduces endogenous D activity [106], and hence should be able to send, but not receive, Ft signal. Such clones induce Vg expression in abutting cells but fail to show a response within the clone (Fig 7D). Hence, the expression of Vg inside *UAS.ft UAS>dpp^{GFP} UAS>wg^{Nrt}* clones, when it occurs, appears to be due to Ft signal coming either from inside or outside the clone.

                                          

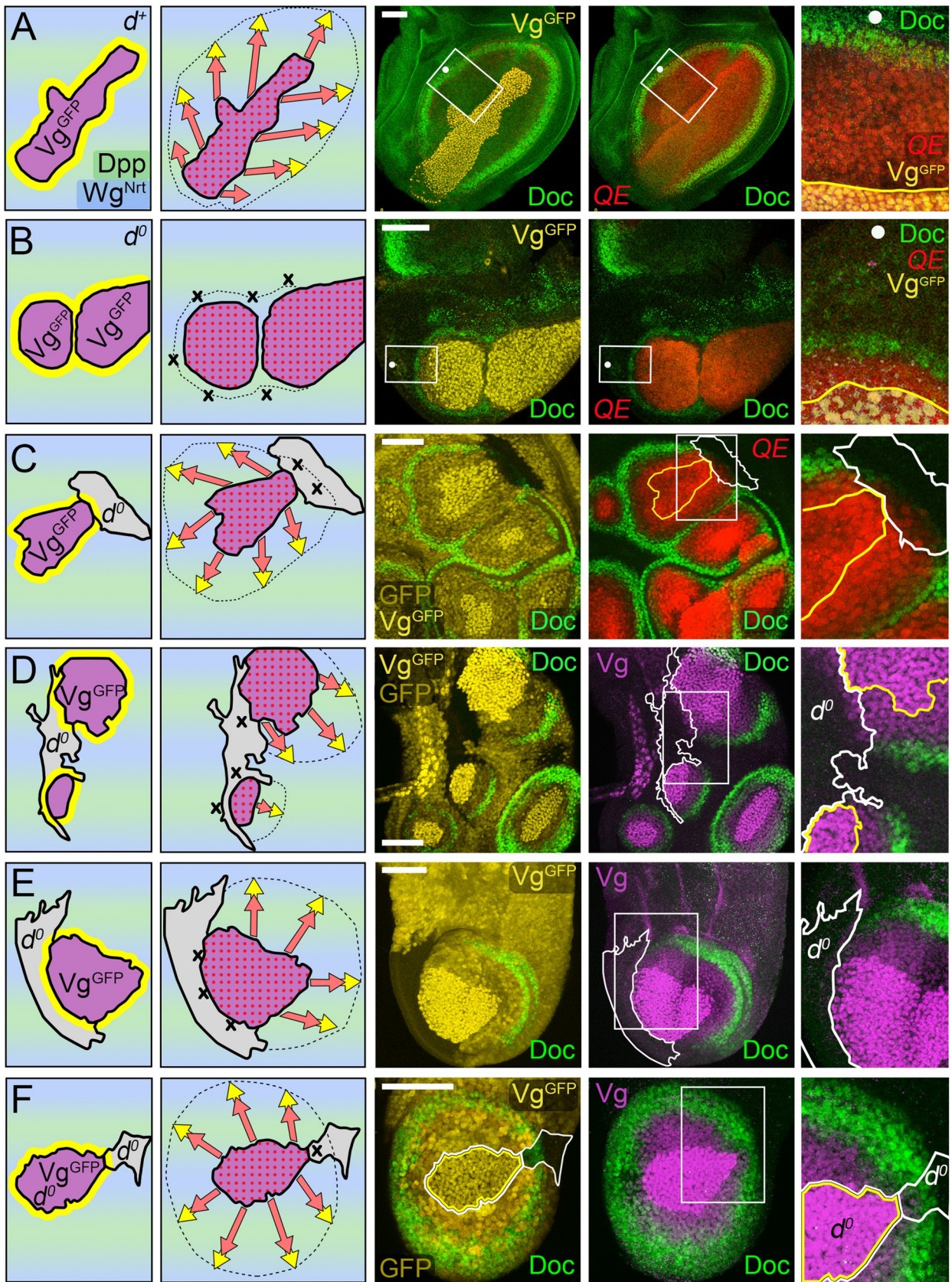

**Fig 8. FF growth in response to Dpp and Wg depends on transduction of Ft signal by D. (A, B)** *Tub>vg*<sup>GFP</sup> clones in *dpp*<sup>d</sup> *ap*<sup>o</sup> *UAS>dpp*<sup>GFP</sup> *UAS>wg*<sup>Nrt</sup> discs that are *wild-type* (*d*<sup>+</sup>, **A**) or null for *d* (*d*<sup>o</sup>; **B**); the clones are shown in purple (for Vg<sup>GFP</sup> expression) in the cartoons to the left and visualized by GFP (yellow) as well as yellow outlining in the insets of the micrographs to the right (rotation of each inset indicated by a white dot). The clone in **A** has induced extensive FF growth, monitored by *5XQE.DsRed* expressing cells in the surround (red), as well as the exclusion of Doc (green), shown at higher magnification in the inset. In contrast, the 2 clones in **B** have induced a weak FF response that is closely restricted to neighboring cells (inset). Note that the level of Doc expression is also reduced, possibly because peak Doc expression requires a minimal level of Ft/Ds signaling that is below the level necessary for initiation of the Vg autoregulatory circuit mediated by the QE. **(C, E)** Discs containing *Tub>vg*<sup>GFP</sup> clones that abut *d*<sup>o</sup> clones (the latter outlined in white in the micrographs). The *d*<sup>o</sup> clones are marked negatively, by the absence of expression of a lineage marker *hsp70.GFP* transgene, in contrast to the *Tub>vg*<sup>GFP</sup> clones, which are marked positively by Vg<sup>GFP</sup> expression. Both GFP and Vg<sup>GFP</sup> are visualized by GFP fluorescence (yellow); hence, clones that are *d*<sup>o</sup> fail to express either and appear black, while *Tub>vg*<sup>GFP</sup> clones express both and appear bright yellow relative to otherwise *wild-type* cells in the surround (which express only GFP and appear dull yellow). Vg<sup>GFP</sup> is also monitored by anti-Vg staining (purple in **D, E**), which appears bright purple in *Tubα1>vg*<sup>GFP</sup> clones owing to the superimposition of Vg<sup>GFP</sup> from the transgene and peak expression of native Vg from the endogenous gene. FF growth of *d*<sup>o</sup> cells that abut *Tubα1>vg*<sup>GFP</sup> clones is severely limited compared to the induction of FF growth in neighboring, otherwise *wild-type* cells, as shown by the greatly reduced range and levels of *5XQE.DsRed* (**C**, red) and native Vg (**D, E**, purple) expression in the magnified insets to the right (as in **B**, Doc expression, green, is also reduced in the *d*<sup>o</sup> cells). **(F)** A *Tub>vg*<sup>GFP</sup> *d*<sup>o</sup> clone (outlined in both white and yellow) that abuts a small *d*<sup>o</sup> clone (outlined in white). The *Tub>vg*<sup>GFP</sup> *d*<sup>o</sup> clone is marked by the superimposition of Vg<sup>GFP</sup> produced by the transgene and native Vg derived from the endogenous gene (bright purple) and has induced FF growth of the surround (visualized by the expression of native Vg, dull purple, as well as the exclusion of Doc, green). Hence, the ability of Dpp plus Wg to induce FF growth depends on D activity in the "receiving" cells (**C, E**), but not in the "sending" cells (**F**). As in **C, E**, both Vg<sup>GFP</sup> and GFP are visualized by innate GFP fluorescence (yellow). However, unlike in **C, E**, the *Tub>vg*<sup>GFP</sup> clone in **F** is null for both *d* and the *hsp70.GFP* transgene. As a consequence, it expresses only Vg<sup>GFP</sup> and appears dull yellow like the surrounding cells, which express only GFP from the *hsp70.GFP* transgene. Taken together with the lack of GFP staining in the neighboring *d*<sup>o</sup> clone, the moderate GFP signal observed in the *Tub>vg*<sup>GFP</sup> *d*<sup>o</sup> clone validates the use of the bright GFP signaling to mark the *Tub>vg*<sup>GFP</sup> clones in **C, E**. Note that for **E** and **F**, analysis was facilitated by using the Minute technique [99] to confer a growth advantage to the *d*<sup>o</sup> clones. **Key genotypes:** *dpp*<sup>d</sup> *ap*<sup>o</sup> *UAS>dpp*<sup>GFP</sup> *UAS>wg*<sup>Nrt</sup> *C765.Gal4 Tub>stop>vg*<sup>GFP</sup> discs that are *d*<sup>+</sup>/*d*<sup>+</sup> (**A**), or *d*<sup>o</sup>/ *d*<sup>o</sup> (**B**); *dpp*<sup>d</sup> *ap*<sup>o</sup> *UAS.dpp UAS>wg*<sup>Nrt</sup> *C765.Gal4 Tub>stop>vg*<sup>GFP</sup> discs that are either *d*<sup>o</sup>*FRT39/d*<sup>+</sup> *hsp70.GFP FRT39* (**C, D**), or *d*<sup>o</sup> *FRT39/d*<sup>+</sup> *hsp70.GFP M(2)25A FRT39* (**E, F**). *ap, apterous*; D, Dachs; Dpp, Decapentaplegic; FF, feed-forward; FRT, >, Flp recombinase targets; QE, Quadrant enhancer; *Tub, Tubulinα1*; Vg, Vestigial; Wg, Wingless; Wg<sup>Nrt</sup>, Neurotactin-Wingless.

In sum, these experiments show that FF growth induced by the combined inputs of Dpp and Wg depends on Ft signal sent by wing cells and transduced in pre-wing cells via the D/Wts/Yki pathway.

## Dpp and Wg are required for FF growth resulting from constitutive activation of the Ft/Ds signal transduction pathway

A central tenet of the FF model is that pre-wing cells must receive all 3 signals, Dpp, Wg, and Ft, to grow and initiate Vg expression, after which they require only Dpp and Wg to maintain Vg expression and grow as wing cells [66]. Accordingly, constitutive activation of the FF transduction pathway should render *dpp*<sup>d</sup> *ap*<sup>o</sup> discs dependent on the joint provision of Dpp and Wg to initiate and sustain QE-dependent wing growth. To test this, we took advantage of the fact that in addition to constituting the FF signal made by wing cells, Ft has a second and distinct role in pre-wing cells, where it functions to block the FF transduction pathway in the absence of the incoming Ft signal (by suppressing D activity and thereby allowing Wts to inhibit Yki activity [66,89]). Hence, one can constitutively activate the FF transduction pathway in *dpp*<sup>d</sup> *ap*<sup>o</sup> discs (where it would otherwise be inactive) by genetically ablating Ft. Accordingly, we have asked whether triply mutant *dpp*<sup>d</sup> *ft*<sup>o</sup> *ap*<sup>o</sup> discs require both Wg and Dpp to initiate and sustain QE-dependent wing growth. As we document in Fig 9, this is indeed the case, and the same is also true when we constitutively activate the FF pathway by other means, either by genetically ablating Wts or over-expressing Yki (S2 and S3 Figs).

In the absence of exogenous Dpp and Wg<sup>Nrt</sup>, *dpp*<sup>d</sup> *ap*<sup>o</sup> discs that are *ft*<sup>o</sup> appear similar to *dpp*<sup>d</sup> *ap*<sup>o</sup> discs: Despite constitutive Yki activity, they fail to express either Vg or *5XQE.DsRed* and form only a rudimentary pre-wing domain (Fig 9A).

To test the requirement for Wg, we generated *dpp*<sup>d</sup> *ft*<sup>o</sup> *ap*<sup>o</sup> *UAS.dpp* discs, which express Dpp uniformly, and asked if they depend on Wg supplied by the induction of *UAS>wg*<sup>Nrt</sup> clones to initiate and sustain wing growth. Unlike *dpp*<sup>d</sup> *ft*<sup>o</sup> *ap*<sup>o</sup> discs, *dpp*<sup>d</sup> *ft*<sup>o</sup> *ap*<sup>o</sup> *UAS.dpp* discs

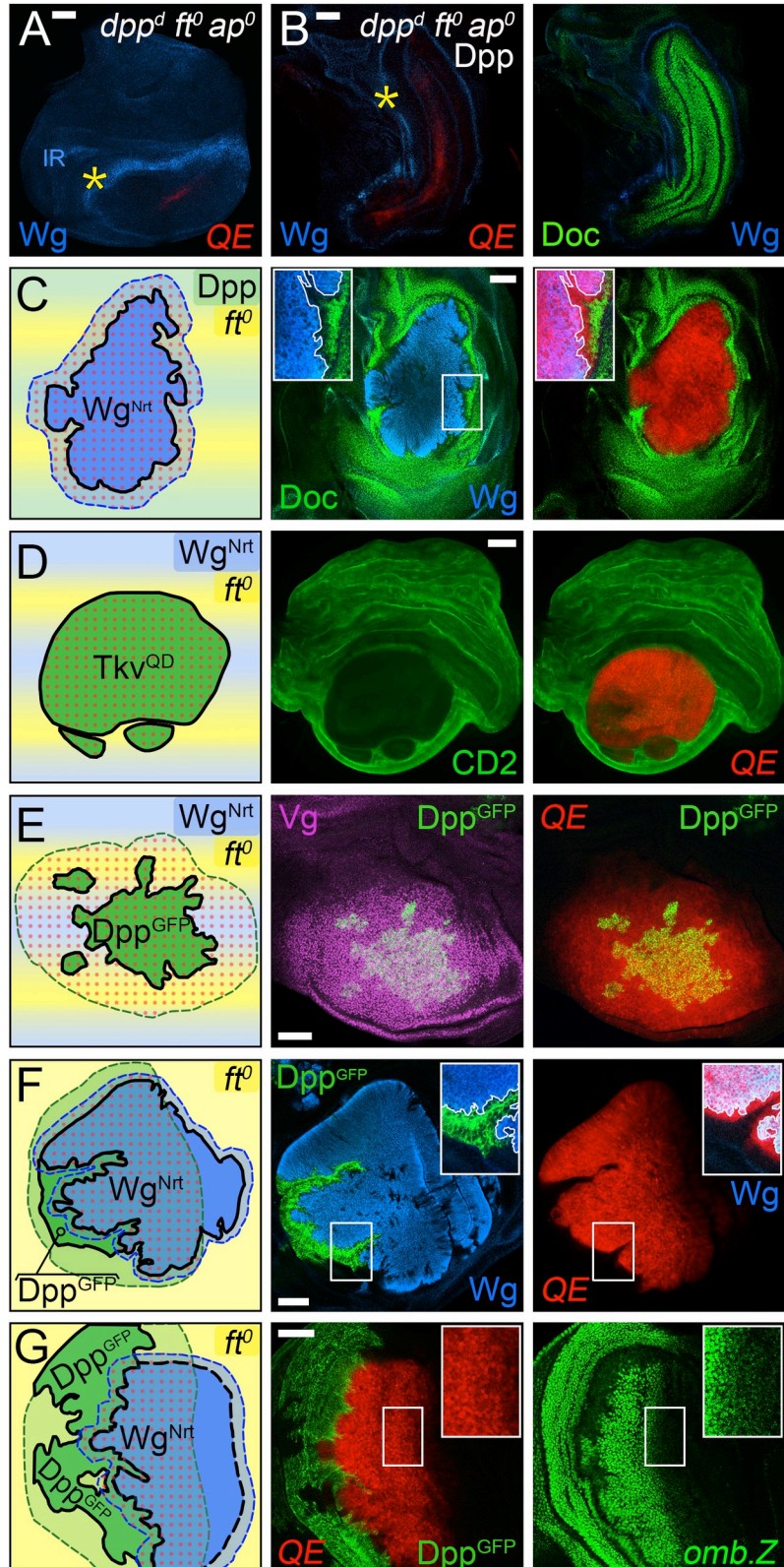

**Fig 9. Dpp is required together with Wg to induce and sustain FF growth in *ft^o* discs. (A, B)** *dpp^d ft^o ap^o* discs that lack (**A**) or uniformly express (**B**) *UAS.dpp*, monitored for *5XQE.DsRed* (red), Wg (blue), and Doc (green; **B** only). The *dpp^d ft^o ap^o* disc forms a greatly reduced pre-wing (delineated by Wg-IR expression in the surrounding hinge, yellow

asterisk), which is virtually devoid of *5XQE.DsRed* expression despite the constitutive activity of the FF transduction pathway (**A**). Supplying such discs with ubiquitous Dpp greatly expands the pre-wing domain, but the cells within express barely detectable levels of *5XQE.DsRed* and high levels of Doc, as also observed for *ftᵒ apᵒ* discs, which retain endogenous Dpp signaling from the A/P boundary (Fig 10D). We interpret these cells as corresponding to *wild-type* pre-wing cells positioned in front of the FF recruitment interface and in the vicinity of the A/P boundary, which receive high levels of Dpp and FF input but only cryptic levels of Wg. We infer such cells cannot up-regulate QE-dependent Vg expression sufficiently to engage the cell-autonomous *vg* autoregulatory circuit, exclude Doc expression and enter the wing proper (see text and Fig 10). (**C–G**) *dppᵈ ftᵒ apᵒ* discs supplied with Dpp/Tkv (green) and Wg^Nrt (blue) input, whether ubiquitously or in marked clones (constitutive activity of the FF transduction pathway is depicted as a yellow wash in the diagrams to the left; red dotted grid indicates QE activity; for simplicity, only the *ftᵒ* genotype, and not the full *dppᵈ ftᵒ apᵒ* genotype, is indicated in the left panels). (**C**) *UAS.dpp* disc containing a *UAS>wg^Nrt* clone (marked by Wg^Nrt expression, blue). In contrast to the disc shown in **B**, all cells within and immediately surrounding the clone express peak levels of *5XQE.DsRed* expression and repress Doc, whereas cells further removed express only high levels of Doc (the inset shows a thin line of the cells abutting the clone that express *5XQE.DsRed* but neither Doc nor Wg^Nrt). (**D, E**) *dppᵈ ftᵒ apᵒ UAS>wg^Nrt* discs carrying either *UAS>tkv^QD* clones (**D**; marked "black" by the absence of expression of a *Tub>CD2* lineage marker, green) or *UAS>dpp^GFP* clones (**E**; marked by Dpp^GFP, green). *UAS>tkv^QD* clones cell-autonomously express peak levels of *5XQE.DsRed* in contrast to *UAS>dpp^GFP* clones, which express peak levels of *5XQE.DsRed* and also induce surrounding cells up to at least 30 cell diameters away to do the same. (**F, G**) *dppᵈ ftᵒ apᵒ* discs carrying abutting clones, 1 *UAS>dpp^GFP* and the other *UAS>wg^Nrt*. The *UAS>dpp^GFP* clones have induced dramatic FF wing growth that extends through most of the *UAS>wg^Nrt* clones, reflecting the extended range of Dpp spread (corroborated in **G** by expression of the *omb.lacZ* reporter, green; the *omb.lacZ* stained image in the inset is overexposed approximately 3-fold to help visualize the fade out in Dpp-dependent *omb.lacZ* expression relative to the extent of *5XQE.DsRed* expression). In contrast, the *UAS>wg^Nrt* clones reciprocally induce only the abutting cells in the *UAS>dpp^GFP* clone to express *5XQE.DsRed*, indicating the tightly limited range of Wg^Nrt (inset in **F**, clone border is outlined in white). Similar results were obtained when the FF pathway was constitutively activated by the absence of *wts* or the overexpression of a *UAS.yki* transgene (S2 and S3 Figs). **Key genotypes:** *dppᵈ ftᵒ apᵒ C765.Gal4* discs that are otherwise *wild-type* (**A**), or *UAS.dpp* (**B**), or *UAS.dpp UAS>stop>wg^Nrt* (**C**), or *UAS>stop>tkv^QD UAS>wg^Nrt* (**D**), or *UAS>stop>dpp^GFP UAS>wg^Nrt* (**E**), or *UAS>stop>dpp^GFP UAS>stop>wg^Nrt* (**F, G**). A/P, anterior–posterior; *ap*, *apterous*; Dpp, Decapentaplegic; FF, feed-forward; FRT, >, Flp recombinase targets; IR, inner ring; QE, Quadrant enhancer; Tkv, Thickveins; *Tub*, *Tubulinα1*; Vg, Vestigial; Wg, Wingless; Wg^Nrt, Neurotactin-Wingless.

(Fig 9B) form an enlarged pre-wing region that expresses very low levels of the *5XQE.DsRed* reporter and uniformly high levels of Doc (the latter indicating high-level Dpp signaling in pre-wing cells that express little or no Vg; indeed, we can barely detect Vg by antibody staining in these discs). As we document in the next section (Fig 10), this response appears to be due to a cryptic supply of Wg that can support both the growth of pre-wing cells as well as barely detectable activity of the QE in response to Dpp but is inadequate to initiate and sustain the cell-autonomous circuit of Vg autoregulation that is required to convert them into wing cells. In contrast, *UAS>wg^Nrt* clones generated in such discs grow as ectopic wing primordia that express uniformly high levels of QE-dependent Vg and induce abutting cells to do likewise, while excluding Doc expression (Fig 9C).

To test the requirement for Dpp, we performed the reciprocal experiment of generating either *UAS>tkv^QD* or *UAS>dpp^GFP* clones in *dppᵈ ftᵒ apᵒ* discs that uniformly express Wg^Nrt. In the absence of exogenous Tkv^QD or Dpp^GFP, *dppᵈ ftᵒ apᵒ UAS>wg^Nrt* discs appear indistinguishable from *dppᵈ ftᵒ apᵒ* discs, forming only a rudimentary pre-wing domain that is devoid of detectable *5XQE.DsRed* or Vg expression. In contrast, *UAS>tkv^QD* clones cell-autonomously activate both outputs and grow as ectopic wing tissue (Fig 9D) and *UAS>dpp^GFP* clones do the same and also induce FF wing growth in the surround that can extend up to approximately 20 to 30 cell diameters away (Fig 9E).

Thus, under conditions in which Dpp and FF input are uniformly high (*dppᵈ ftᵒ apᵒ UAS.dpp* discs), FF growth appears to depend on the expression and range of Wg signaling, and the same is the case for Dpp in reciprocal experiments in which Wg and FF input are uniformly high (*dppᵈ ftᵒ apᵒ UAS>wg^Nrt* discs).

To further assess these findings, we co-induced *UAS>dpp^GFP* and *UAS>wg^Nrt* clones in *dppᵈ ftᵒ apᵒ* discs. Under these conditions, *UAS>dpp^GFP* clones can induce FF wing growth

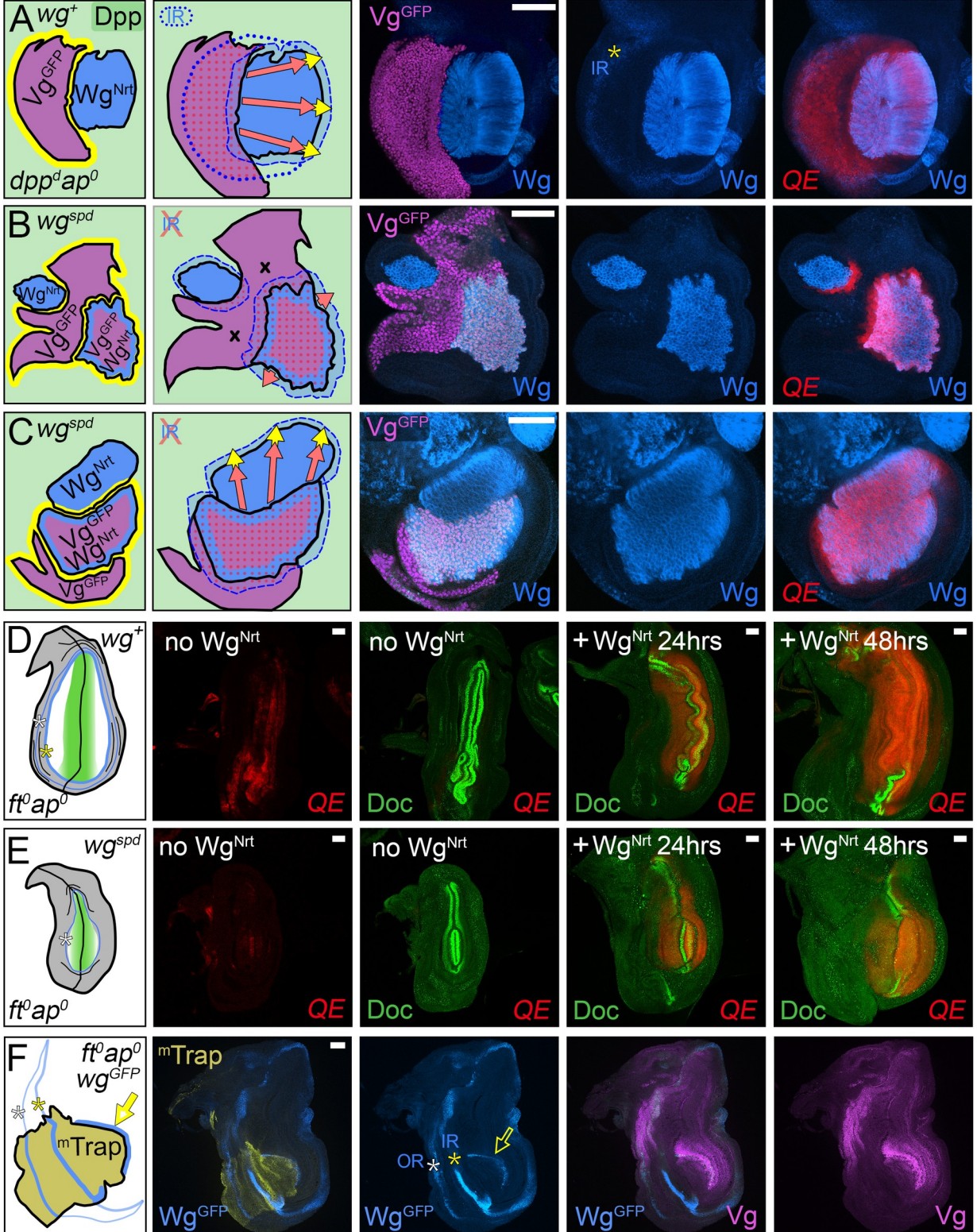

**Fig 10. Source and contribution of "cryptic" Wg in *ap* null discs. (A–C)** $dpp^d$ $ap^o$ *UAS.dpp* discs that are otherwise *wild-type* (**A**) or homozygous for the $wg^{spd}$ allele (**B, C**) and carry abutting $Tub>vg^{GFP}$ (purple), $UAS>wg^{Nrt}$ (blue), and $Tub>vg^{GFP}$ $UAS>wg^{Nrt}$ (pinkish blue) clones. Wg$^{Nrt}$ is visualized using an anti-Wg antisera, which also shows the Wg-IR (yellow asterisk) in the prospective hinge (**A**; absent in **B, C**). In the presence of Wg-IR expression (**A**), all cells within the $Tub>vg^{GFP}$ clone express high levels of the *5XQE.DsRed* (red) even when located

many cell diameters away from the abutting $UAS>wg^{Nrt}$ clone. This output appears to be due to "cryptic" Wg made by the IR cells, as it does not occur in discs that lack this source of Wg (**B**). Instead, the $Tub>vg^{GFP}$ clones in this disc do not express $5XQE.DsRed$, except in cells that contact the abutting $UAS>wg^{Nrt}$ and $Tub>vg^{GFP}$ $UAS>wg^{Nrt}$ clones [we infer that the sole $UAS>wg^{Nrt}$ clone in **B** (upper left) is located in the hinge region adjacent to the pre-wing territory and hence can send Wg^Nrt signal to the abutting $Tub>vg^{GFP}$ cells in the pre-wing domain but cannot respond to FF signal sent from the $Tub>vg^{GFP}$ clone]. The discs in (**A**) and (**C**) confirm that clones that express the $Tub>vg^{GFP}$ transgene induce extensive FF growth in abutting $UAS>wg^{Nrt}$ clones, irrespective of whether the disc is $wg^+$ or $wg^{spd}$. (**D, E**) $5XQE.DsRed$ and Doc expression in $ft^o ap^o$ discs that are $wg^+$ (**D**) or $wg^{spd}$ (**E**). The left-most micrographs are overexposed approximately 3-fold for $5XQE.DsRed$ signal to visualize otherwise cryptic expression; the remaining micrographs to the right are double stained for $5XQE.DsRed$ and Doc expression under standard conditions. In the $wg^+$ discs, the pre-wing territory is rudimentary and expresses low levels $5XQE.DsRed$ and high levels of Doc. We infer that cells in this domain have received high levels of Dpp but are unable to up-regulate the QE-dependent Vg expression despite constitutive activation of the FF transduction pathway because of the limited amount of "cryptic" Wg. However, their capacity to do so is rapidly restored by exogenous Wg generated throughout the disc either 24 or 48 hours before assaying for $5XQE.DsRed$ and Doc expression. In the $wg^{spd}$ discs, $5XQE.DsRed$ expression is virtually undetectable in the pre-wing territory. However, as in $wg^+$ discs (**D**), high-level $5XQE.DsRed$ expression and growth are restored by the late provision of exogenous Wg. (**F**) A morphotrap (^mTrap) expressing clone (gold) in a $ft^o ap^o$ disc that carries a fully functional, $wg^{GFP}$ knock-in allele in trans to the *wild-type* allele. The clone includes cells in the Wg-IR and Wg-OR of the prospective hinge (yellow and white asterisks), where Wg^GFP (blue) accumulates to abnormally high level in the cells that express it, confirming entrapment of Wg^GFP expressed by these cells. It also contributes to the pre-wing territory, where Wg^GFP accumulates selectively in ^mTrap expressing cells at the clone border (yellow arrow). These same cells, and flanking cells on either side also up-regulate native Vg (purple) and grow as wing tissue (see text; note that Vg is also expressed in a proximal portion of the prospective hinge domain; this expression is not dependent on the QE and unrelated to FF signaling in the pre-wing and wing territories, as in Fig 2E). **Key genotypes: (A–C)**: $Tub>stop>vg^{GFP}$ $UAS>stop>wg^{Nrt}$ *UAS.dpp* *C765.Gal4* in either $dpp^d wg^+ ap^o$ discs (**A**) or $dpp^d wg^{spd} ap^o$ discs (**B, C**). (**D, E**) $UAS>wg^{Nrt}$ *Tub.Gal80^ts* *C765.Gal4* discs that are either $ft^o wg^+ ap^o$ (**D**) or $ft^o wg^{spd} ap^o$ (**E**). (**F**) A $ft^o wg^{GFP} ap^o/ ft^o wg^+ ap^o$ disc that is *rn.Gal4* $Tub>Gal80 stop>Gal4$ $UAS>$^mTrap. *ap*, apterous; Dpp, Decapentaplegic; FF, feed-forward; FRT, >, Flp recombinase targets; Ft, Fat; IR, inner ring; ^mTrap, Morphotrap; OR, outer ring; QE, Quadrant enhancer; *Tub*, *Tubulinα1*; Vg, Vestigial; Wg, Wingless; Wg^Nrt, Neurotactin-Wingless.

that extends as many as 30 to 40 cell diameters into neighboring $UAS>wg^{Nrt}$ clones, as monitored by $5XQE.DsRed$ expression, whereas $UAS>wg^{Nrt}$ clones can only induce abutting cells to express the $5XQE.DsRed$ reporter (Fig 9F and 9G). Notably, the extent of FF growth observed in such $UAS>wg^{Nrt}$ clones appears to depend on the range of Dpp^GFP spreading from the $UAS>dpp^{GFP}$ clones, as corroborated by monitoring expression of an *omb.lacZ* reporter, which is activated by moderate to high levels of Dpp input (Fig 9G). Conversely, FF growth induced by $UAS>wg^{Nrt}$ clones is severely restricted by limited range of the Wg^Nrt signal.

Finally, we performed an additional experiment to corroborate the identification of Ft as the FF signal. To do so, we induced $ft^o$ clones that co-express $UAS>dpp$ in $dpp^d ap^o$ $UAS>wg^{Nrt}$ discs. As expected, such clones constitutively activate the FF transduction pathway and express QE-dependent Vg at peak level. However, they do so in a strictly cell autonomous fashion, despite neighboring cells being in receipt of both Dpp and Wg^Nrt (S4A Fig). This result extends our previous evidence that Ft itself, rather than some other factor induced by peak Vg activity within the clone, is the FF signal [66]. To corroborate this interpretation, we used expression of a $UAS.ds^{RNAi}$ transgene to knock down Ds activity in clones that co-express $UAS>dpp^{GFP}$ and $UAS>wg^{Nrt}$ in $dpp^d ap^o$ discs. Knock-down of Ds, like the removal of Ft, constitutively activates the FF signal transduction pathway. Accordingly, $UAS.ds^{RNAi}$ $UAS>dpp^{GFP}$ $UAS>wg^{Nrt}$ clones in $dpp^d ap^o$ discs activate QE-dependent Vg expression cell-autonomously. However, in contrast to $ft^o$ clones, these $UAS.ds^{RNAi}$ clones retain the capacity to generate FF signal and act non-autonomously to induce QE-dependent wing growth in the surround (S3B Fig).

Taken together, these results establish (i) that Ft, itself, is the FF signal responsible for inducing Dpp and Wg-dependent Vg expression in abutting cells; and (ii) that under conditions in which the FF signaling transduction pathway is constitutively active, QE-dependent wing growth depends on the ranges and actions of both Dpp and Wg.

## The source and contribution of "cryptic" Wg to FF growth in *ap* null discs

The FF model stipulates that both the initiation and maintenance of QE-dependent Vg expression should depend, in an absolute manner, on the combined inputs of Dpp, Wg, and Yki/Vg

(Yki being required for initiation and Vg for up-regulation and maintenance) [66]. However, both here (Fig 9B) and previously [64–66]), we have found that low levels of QE activity can be induced and sustained in $ap^o$ discs, either by the ectopic expression of sufficiently high levels of exogenous Vg (e.g., under *Tub* control) or by constitutive activation of Yki (e.g., by the removal of Ft or Wts, or the overexpression of Yki; Fig 9B, S2 and S3 Figs)—even though $ap^o$ discs lack D/V border cells, the posited source of Wg upon which FF wing growth depends. These findings indicate either that Wg is not an absolute requirement for QE activity, challenging the FF model, or alternatively, that there is a source of "cryptic" Wg in $ap^o$ discs. To distinguish these possibilities, we have asked if we can identify such a source, and if so, determine whether it is responsible for the unexpected QE response.

In *wild-type* wing discs, Wg is expressed not only in D/V border cells, but also in cells outside of the pre-wing territory, most notably in an "inner ring" (IR) of cells that surrounds the pre-wing territory (Fig 2A and 2C). Although the IR is greatly reduced in diameter in $ap^o$ discs, it still surrounds what remains of the pre-wing territory (Fig 2A and 2F). To assess if the IR is the source of cryptic Wg in $ap^o$ discs, we have performed 3 sets of experiments.

The first 2 sets of experiments take advantage of a *wg* enhancer-deletion allele, $wg^{spade-flag}$ ($wg^{spd}$), which fails, selectively, to express *wg* in IR cells [108]. In the first set, we focused on the unexpected observation that ectopic Vg-expressing clones can cell-autonomously activate QE reporter transgenes in $ap^o$ discs, despite the absence of Wg produced by D/V boundary cells [64–66]. We first corroborated this observation by co-inducing $Tub>vg^{GFP}$ and $UAS>wg^{Nrt}$ clones in $dpp^d\ ap^o\ UAS.dpp$ discs that are *wild-type* for *wg*, a condition in which Dpp is not limiting and the level of ectopic Vg$^{GFP}$ is similar to that of peak, endogenous Vg. As expected based on our prior findings, $Tub>vg^{GFP}$ clones generated in such discs grow as wing tissue and uniformly express readily detectable levels of the *5XQE.DsRed* reporter (Fig 10A). Moreover, they can induce similar levels of *5XQE.DsRed* expression, as well as FF growth, in abutting $UAS>wg^{Nrt}$ clones (Fig 10A). To assess whether the cell-autonomous *5XQE.DsRed* expression observed in $Tub>vg^{GFP}$ clones depends on Wg coming from IR cells, we repeated this experiment, except in discs that are $wg^{spd}$ rather than $wg^+$. In this case, we find that $Tub>vg^{GFP}$ clones do not express *5XQE.DsRed*, nor do they induce surrounding cells to do so, unless they co-express the $UAS>wg^{Nrt}$ transgene or abut cells that do (Fig 10B and 10C).

Hence, the $wg^{spd}$ allele greatly reduces or eliminates a source of cryptic Wg that is otherwise responsible for the unexpected QE activity in $Tub>vg^{GFP}$ clones in $dpp^d\ ap^o\ UAS.dpp$ discs.

In the second series of experiments, we focused on the unexpected expression of QE transgenes in $ap^o\ ft^o$ discs ([66]; corroborated in $dpp^d\ ft^o\ ap^o\ UAS.dpp$ discs; Fig 9B), which are similarly devoid of Wg that would normally be produced by D/V border cells. In $ft^o\ ap^o$ discs, Dpp is provided by endogenous *dpp* expression along the A/P compartment boundary, and Yki is constitutively active owing to the absence of Ft. When these discs are $wg^+$, we observe low level *5XQE.DsRed* expression in an extended stripe of cells that co-expresses high levels of Doc (Fig 10D; here and in Fig 10E, the signal in the QE-only micrographs has been overexposed to visualize weak *5XQE.DsRed* expression). As above (Fig 9B), we interpret these Doc expressing cells as corresponding to pre-wing cells that are beginning to activate the QE in response to a cryptic supply of Wg, but have not received sufficient Wg to drive the Vg autoregulatory circuit and suppress Dpp-dependent Doc expression. By contrast, in $ft^o\ ap^o$ discs that are $wg^{spd}$, both *5XQE.DsRed* expression as well as the Doc-expressing pre-wing territory is greatly reduced, consistent with a corresponding reduction in cryptic Wg (Fig 10E). Corroborating this interpretation, we find that providing such discs with uniform Wg$^{Nrt}$ (under $Gal80^{ts}/Gal4$ control) during the last 24 or 48 hours of larval development is sufficient to initiate FF growth of wing tissue expressing high levels of *5XQE.DsRed* expression, centered on the A/P compartment

boundary (Doc expression being suppressed by endogenous Vg [101], which is co-expressed with the reporter).

In the third series of experiments, we asked whether we could use morphotrap technology [105] to concentrate and visualize signaling by otherwise cryptic Wg in $ft^o\ ap^o$ discs. To do so, we generated $UAS>^mTrap$ expressing clones in $ft^o\ ap^o$ discs that are also heterozygous for a fully functional, GFP knock-in allele of wg ($wg^{GFP}$ [109]). As illustrated in the example shown in Fig 10F, $UAS>^mTrap$ clones that extend across the IR and contribute to the pre-wing territory of $ft^o\ wg^{GFP}\ ap^o/ft^o\ wg^+\ ap^o$ discs accumulate readily detectable levels of $Wg^{GFP}$ at the interface between the clone and the abutting non-$^mTrap$ expressing cells within the pre-wing domain. The abnormal accumulation along the clonal interface is also associated with the local induction of readily detectable Vg expression, an output not otherwise observed in the pre-wing territory of $ft^o\ ap^o$ discs. Given that high-level $^mTrap$ expression within the clone would be expected to sequester any $Wg^{GFP}$ made by the $^mTrap$ expressing cells themselves (corroborated by the strong local $Wg^{GFP}$ accumulation associated with IR and OR cells within the clone), we infer that the local accumulation of $Wg^{GFP}$ at the clonal interface reflects the trapping of $Wg^{GFP}$ that has moved through the pre-wing epithelium until it is sequestered by $^mTrap$ expressing cells at the clone border. Given the results of the first 2 sets of experiments, we consider IR cells to be the most likely source of this cryptic Wg.

In sum, we conclude that the unexpected induction of QE activity caused by ectopic expression of Vg or constitutive activation of Yki in $ap^o$ discs is due to a cryptic supply of Wg, most likely provided by IR cells surrounding the pre-wing domain. Importantly, this level of cryptic Wg is not sufficient to allow $Tub>vg$ clones to induce the QE-dependent Vg autoregulatory circuit in surrounding cells, nor to allow constitutive activity of Yki to cell-autonomously initiate the circuit. Hence, we infer that it is normally of little biological consequence for wing growth, a conclusion supported by the $wg^{spd}$ mutant phenotype in which the wing is close to normal size despite the absence of Wg that would otherwise be produced by IR cells [108, 170].

## Morphogen is required for growth as well as specification of wing cells

A central element of the FF model is that Vg expression not only defines cells as belonging to the wing, but also programs them to gain mass and proliferate as long as they receive minimum thresholds of Dpp and Wg. The preceding experiments establish that Dpp and Wg act to recruit and retain cells in the growing wing primordium by fueling QE-dependent Vg expression. They do not, however, provide evidence that both morphogens are also required for the survival and growth of wing cells once they are defined as such by Vg.

Previously, we tested this for Wg by generating wing cells that express Vg constitutively under the control of the $Tub$ promoter, thus bypassing the normal requirement for Wg to maintain QE-dependent Vg expression, and then asked if these cells still require Wg to survive and grow [65]. Specifically, we co-induced clones of cells that are null for the Wg co-receptor Arrow (Arr; *Drosophila* Low-density lipoprotein Receptor-related Protein 6, LRP6; [110]) together with $Tub>vg$ clones in wing discs that carried the *5XQE.DsRed* reporter but were otherwise *wild-type*. All such $arr^o$ clones located within the prospective wing divide a few times (likely rescued by transient perdurance of Arr protein) but then lose *5XQE.DsRed* expression and are eliminated from the epithelium, irrespective of whether they do or do not co-express the $Tub>vg$ transgene. These results indicate that Wg is required not only to induce and sustain QE-dependent Vg expression in wing cells, but also to sustain the survival and growth of cells defined as wing by Vg.

To determine if Dpp is similarly required, we have repeated this experiment, only this time co-inducing $Tub>vg^{GFP}$ clones together with clones that are either null for the Dpp receptor

Tkv or null for the native *vg* gene (the latter providing a control for the rescuing capacity of Vg^GFP generated by the transgene). As we observed for *arr^o* clones, *tkv^o* clones in the wing grow transiently, lose *5XQE.DsRed* and are eliminated regardless of whether they also carry the *Tub>vg^GFP* transgene (Fig 11A and 11B, note the absence of *tkv^o*/*tkv^o* clones, marked "black" by the elimination of the *arm.lacZ* lineage marker (green) in most or all of the wing primordium, which is demarcated by a dotted white line, irrespective of expression of the *Tub>vg^GFP* transgene, purple; reciprocal *tkv^+*/*tkv^+* sister clones are apparent as bright green patches in the background of otherwise *tkv^o*/*tkv^+* cells, dull green). In contrast, *vg^o* clones that carry the *Tub>vg^GFP* transgene survive and grow as normal wing tissue, whereas clones that do not carry the transgene are rapidly lost from the wing primordium (Fig 11C). Hence, providing exogenous Vg^GFP that is competent to maintain the wing state in the absence of endogenous Vg does not bypass the requirement for cells to transduce Dpp to survive and grow as wing tissue.

Thus, in addition to their co-equal roles in inducing and maintaining QE-dependent *vg* transcription, both Dpp and Wg are required for the sustained survival and growth of wing cells after they are defined as such by Vg.

## Discussion

How morphogens control growth constitutes a fundamental, unsolved problem in animal development. In the case of the *Drosophila* wing, an intensively studied paradigm, most attention has focused on Dpp (a BMP), which is produced by cells along the A/P compartment boundary and spreads outwards forming gradients that extend both anteriorly and posteriorly from the boundary. It is well established that these gradients provide a series of concentration thresholds that organize wing pattern by controlling target gene expression as a function of distance from the source. They also control wing growth, but strikingly, in a manner that does not show a simple relationship between morphogen concentration and the rate of growth of individual cells. Instead, cells throughout the wing appear to gain mass and proliferate uniformly, despite receiving different amounts of Dpp. Some models explain this lack of correspondence by positing that individual cells do not grow in response to the amount of Dpp they receive, but rather to the slope of the Dpp gradient [36–39] or to changes in the amount of Dpp they receive over time [40–42]. Others posit that cells respond to the amount, but in a manner that is balanced out by molecular [43–45] or mechanical feedbacks [57–61], or by the cellular "memory" of prior exposure [13,25,62,63]. However, none of these models are supported by compelling evidence, and no satisfactory consensus has been reached.

Previously, we proposed a different model in which growth is governed by a progressive increase in morphogen range [64–66]. This model was based on our analysis of the role of Wg (a Wnt), another morphogen that controls wing growth, albeit along the dorso/ventral rather than the antero/posterior axis. Specifically, we posited that the outward spread of Wg from source cells along the D/V compartment boundary expands the domains in which Wg can sustain the specification and growth of cells within the wing and induce surrounding pre-wing cells to grow and enter the wing. Further, we provided evidence that it does so by driving the feed-forward (FF) autoregulation of the selector gene *vg*, which defines the wing state and controls the recruitment and growth of wing cells in response to Wg. However, the conjecture that Dpp operates along the antero/posterior axis by the same FF mechanism [64–66] has received little consideration (e.g., [30–32,34,35]). Here, we have subjected Dpp to the same experimental tests on which the FF model for Wg is based and find that Dpp behaves indistinguishably. These findings argue that the FF model applies equally to both Wg and Dpp and provide a unifying explanation for how they act together to control wing growth.

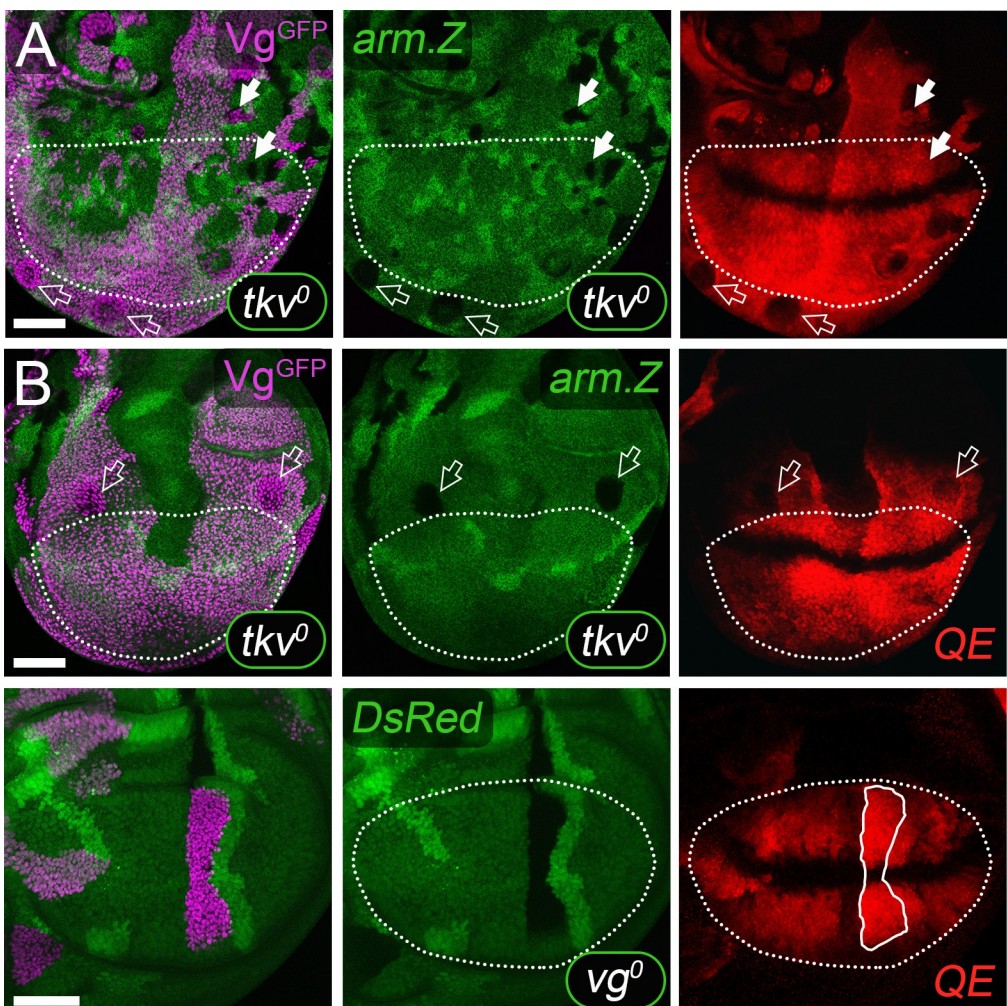

**Fig 11. Dpp is required for the survival and growth of wing cells in addition to their specification by QE-dependent Vg expression.** *Wild-type* wing discs in which either *tkv^o* clones (**A,B**) or *vg^o* clones (**C**) have been co-induced with *Tub>vg^GFP* clones. The *Tub>vg^GFP* clones are marked positively by GFP (purple); the *tkv^o* and *vg^o* clones are marked "black" by the absence of a marker transgene (green; *arm.lacZ* in **A, B**, and *Tub.DsRed* in (**C**), whereas their *wild-type* "twin" clones are homozygous for the marker transgene and appear bright green. (**A, B**) Co-induction of *tkv^o* and *Tub>vg^GFP* clones. Discs were heat shocked during the first instar, and a second time approximately 48 hours (**A**) or approximately 72 hours (**B**) before the end of larval life. Both early and late induced *Tub>vg^GFP* clones survive well throughout the entire disc, the early clones forming large patches that up-regulate the expression of *5XQE. DsRed* in the wing proper (outlined by a white dashed line) as well as in the surrounding pre-wing territory where they cause precocious conversion and expansion of the latter as wing tissue. In contrast, and as previously observed [23], *tkv^o* clones survive and grow only transiently in the wing proper, owing to perdurance of *wild-type* gene product, but are lost after 48 hours of clone induction in the central region (**A**) and after 72 hours at the wing periphery (**B**; the delayed loss of *tkv^o* clones at the lateral wing periphery might reflect the initially higher levels of Tkv expression in these regions, and hence longer perdurance of residual Tkv following clone induction). Importantly, this behavior is observed regardless of whether the *tkv^o* clones carry or lack the *Tub>vg^GFP* transgene, even when they are induced late, within large, early induced *Tub>vg^GFP* clones. The 2 solid white arrows in the upper right of **A** indicate 2 examples of surviving, lateral *tkv^o* clones that still express the *5XQE.DsRed* reporter, one of which belongs to a *Tub>vg^GFP* clone while the other does not (the size and distribution of the sibling "twin spot" clones serve as a positive control for the timing and success of clone induction). *tkv^o* clones located in the abnormally expanded domains of QE-reporter expressing cells outside of the wing proper round up and lose QE reporter gene expression within 48–72 hours of clone induction. The empty white arrows in **A** and in **B** indicate examples of such *tkv^o* clones at 48 and 72 hours after clone induction, respectively. (**C**) Co-induction of *vg^o* and *Tub>vg^GFP* clones. Like *tkv^o* clones, *vg^o* clones survive and grow only transiently within the wing proper, although they survive and grow normally in the more proximal hinge and body wall territories [62]. However, *vg^o Tub>vg^GFP* clones grow normally and express QE reporters in the wing, confirming that *Tub>vg^GFP* transgene encodes functional Vg that can specify the wing state. In the example shown, clones were induced with a single heat shock approximately 96 hours before the end of larval life: The wing proper

carries a single, large $vg^o$ $Tub>vg^{GFP}$ clone, outlined in white, that expresses the *1XQE.lacZ* reporter, adjacent to its *wild-type* twin. A surviving $vg^o$ clone in the hinge/body wall territory that does not co-express the $Tub>vg^{GFP}$ clone is visible at the top of the image. **Key genotypes: (A, B)** $tkv^o$ *FRT40/arm.lacZ FRT40, Tub>stop>vg$^{GFP}$/+*, **(C)** *FRT42 $vg^o$/ FRT42 Tub.DsRed; Tub>stop>vg$^{GFP}$/+*. D/V, dorso–ventral; Dpp, Decapentaplegic; FRT, >, Flp recombinase targets; IR, inner ring; QE, Quadrant enhancer; Tkv, Thickveins; *Tub, Tubulinα1*; Vg, Vestigial.

## The feed-forward model: Control of growth by morphogen range

As outlined in Fig 1, the FF model stipulates that Dpp and Wg drive the major phase of wing growth by fueling 2 distinct circuits of *vg* autoregulation, both mediated by the *vg* Quadrant enhancer (QE). The first operates cell-autonomously in wing cells, via the capacity of Vg to associate directly with the QE and sustain *vg* transcription in response to Dpp and Wg. The second constitutes an intercellular circuit of autoregulation in which *vg* activity in wing cells generates the FF signal, the protocadherin Ft, which acts at short-range on neighboring pre-wing cells to induce them to express *vg* [66]. It, too, depends on the QE, via the capacity of Yki, the transcriptional effector of the Ft signal, to associate with the enhancer and activate *vg* transcription in response to Dpp and Wg [64–66]. Hence, as Vg levels rise in the responding pre-wing cells, they adopt the wing state, and switch to becoming dependent on Vg, rather than Yki, to sustain the cell-autonomous circuit of Vg autoregulation.

Importantly, Dpp and Wg do not control wing growth solely via their capacity to initiate and sustain Vg expression. Instead, once Vg is active, it programs wing cells to grow in response to Dpp and Wg. Indeed, one can bypass the normal requirement for both circuits of Vg autoregulation by supplying exogenous Vg from a constitutively expressed transgene. However, such Vg-expressing wing cells remain absolutely dependent on both Dpp and Wg to survive and grow within the wing primordium (Fig 11; [65]). Accordingly, we infer that as Dpp and Wg spread outwards from the A/P and D/V boundaries, they not only induce and sustain Vg expression, but act through its agency to increase wing mass and cell number. Hence, as previously proposed [64–66; 111], Dpp and Wg act instructively to control wing growth as a function of morphogen range. In agreement, wing growth during larval life appears to cease when Dpp and Wg reach their maximum ranges, but can be reinitiated by manipulations that extend their ranges [171].

## Feed-forward versus morphogen slope and temporal dynamic models

The FF model explains key findings that are not readily compatible with the morphogen slope [36–39] or temporal dynamics [40–42] models. In particular, these models rely on differentials in morphogen abundance, either in space or time, to control the rate at which individual cells grow. Hence, according to such models, experimental interventions that flatten these differentials should terminate growth. However, they do not. Instead, they sustain growth in central portions of the wing and stimulate excess growth at the wing periphery [111–114]. In contrast, the FF model stipulates that growth within the wing proper should depend on cells receiving a minimum threshold of both morphogens, whereas growth and recruitment of surrounding pre-wing cells should be limited by their access to both morphogens. As a consequence, uniform provision of either Dpp or Wg should have little if any effect on growth of the wing proper, provided they exceed the minimum thresholds required. In contrast, they should result in excess growth and recruitment of pre-wing cells at the wing perimeter, where access is otherwise restricted—as is observed.

We note that many previous studies of Dpp draw an unexplained distinction between "medial" and "lateral" wing growth, with medial growth requiring a minimum threshold of Dpp but being refractory to ectopic Dpp in contrast to lateral growth, which is enhanced by ectopic Dpp [24,37,38,52,105,111,113,115]. In our view, these translate to the cell-autonomous

and intercellular modes of FF growth that operate, respectively, in wing cells behind versus pre-wing cells in front of the FF recruitment interface. Accordingly, we posit that "lateral" growth induced by ectopic morphogen is synonymous with constitutive propagation of the intercellular circuit and recruitment of pre-wing cells, which is then sustained by "medial" growth of the recruited wing cells by the cell-autonomous circuit.

## Feed-back versus feed-forward models of wing growth

Unlike slope and temporal dynamic models, other models that accept the premise that growth is controlled by morphogen grade explain the uniform growth of the wing by invoking mechanical or molecular feed-backs that "equalize" the response of wing cells that receive different concentrations depending on their positions (reviewed in [29–35]). For mechanical feedback models, it has been proposed that the progressive expansion of the wing is physically constrained by the rigidity of the surrounding prospective hinge and body wall tissue, leading to the compaction of centrally located cells and their reduced growth in response to high levels of Dpp and Wg [57–61]. However, a causal link between mechanical tension and wing growth has been challenged by in vivo experiments that show few if any detectable effects on growth following removal of the physical constraints imposed by surrounding tissues [116]. Our results likewise argue against such a causal relationship by providing evidence that wing growth is governed predominantly by morphogen range and propagation of the FF recruitment circuit even under conditions in which the surrounding cells should impose the same mechanical constraints as in normal development.

In contrast, there is ample evidence for molecular feedbacks. In particular, both Dpp and Wg regulate the expression of their receptors, other cell surface–binding proteins, and transcriptional effectors to create feedback circuits that modulate morphogen spread as well as the transcriptional responses of individual cells to their signaling activities (reviewed in [43–46]). For example, Dpp regulates the expression of its receptor Thickveins (Tkv; [102,115]), the Dpp-binding glypican Dally [117–121], a secreted protein, Pentagon (Pent; [122,123]), that regulates Dally [122,124], and the transcription factors Brinker (Brk; [48,55,56,113]) and Daughters against Dpp (Dad; [47,51]), an inhibitor of one of its primary transcriptional effectors, Mothers against Dpp (Mad). Similarly, Wg regulates the expression of two of its co-receptors, Frizzled2 (Fz2; [125,126]) and Arrow [110,126] as well as a non-transducing decoy receptor, Frizzled3 (Fz3) [127,128,126], and the related glypicans Dally and Dally-like (which also bind Wg; [129–133]). We favor the view that the primary feedbacks that govern wing growth are those that control morphogen range, e.g., by modulating the abundance of receptors and other cell surface–binding proteins. This possibility does not negate a role for feedbacks that modulate target gene expression and patterning in response to the amount of morphogen individual cells receive, but relegates them to a subsidiary role in the control of growth.

We envisage that in addition to its other roles in specifying the wing state, Vg controls how the feedback circuits that modulate morphogen range operate within the developing wing. Accordingly, by inducing and sustaining Vg expression, Dpp and Wg may act through the agency of Vg to achieve an additional layer of FF regulation that governs their own ability to spread and control growth.

## Dpp versus Wg: Identical roles but different views on the means and necessity of morphogen spread

Morphogens, by definition, exert their global control by moving away from producing cells and acting directly and at long range on surrounding tissue. That Dpp fulfills this key criterion in the wing is manifest by experiments that confirm the strict confinement of its production to

A compartment cells and its direct and long-range actions on the adjoining P compartment [16,17,105,111,112,134,135]. However, no such compartment-based segregation occurs between producing and responding cells in the case of Wg. Instead, following D/V compartmentalization, most of the nascent wing cells on both sides of the boundary are induced to express Wg as well as Vg under DSL/Notch control (Fig 1C; [13,63,80]). It is only during subsequent development, as Wg production remains confined to cells close to boundary and the wing continues to expand that the FF model requires Wg to act directly and at long range (Fig 1D)—a possibility that remains open to challenge [32,35,63,136].

Most prominently, gene editing experiments in which native Wg is replaced by a membrane tethered form of Wg (Neurotactin-Wg, Wg$^{Nrt}$), have revealed a remarkable, albeit still compromised, capacity of Wg to sustain wing growth even when it appears unable to disengage from producing cells [63]. Although there are several explanations of the rescuing capacity of Wg$^{Nrt}$ that do not contravene a normal requirement for long-range action of native Wg [137], these and other experiments have been presented as evidence against such long-range action, and instead, in support of a "cellular memory" model [13,62,63]. According to this model, transient Wg signaling induces most if not all cells in the nascent wing to adopt a heritable state that directs their descendants to continue growing even when they are no longer in range to receive Wg. Based on our present results as well as prior findings, we regard this interpretation, as well as the challenge it presents for the FF model, as untenable.

First, we provide matched experiments in which we compare tethered versus "free" forms for both Dpp and Wg. Reinforcing previous findings with Wg$^{Nrt}$ [18,64,66] and morphotrapped Dpp [105]; we show that tethered forms of both morphogens support only local growth, limited to a few to several cell diameters from producing cells. In contrast, "free" forms can fuel growth extending up to 30 or more cell diameters away—similar to what happens during normal wing development. These results contradict the cellular memory model. Second, this model is also contradicted by independent, and in our view compelling, experiments that show that cells located at the wing periphery, far from the normal sources of Dpp and Wg, require continuous, direct input from both morphogens to sustain *vg* expression and to grow ([18,19,23,24,65,138,170]; see also Fig 11). Finally, studies of tethered forms of Wg and other Wnts in other contexts confirm that Wnts can, and sometimes must, act as long-range organizing signals [139–142].

Recent challenges to the long-range action of Wg [63,136] have amplified a long held bias toward Dpp as the prime if not sole morphogen governing wing growth [29–35]. Our present results confirm and extend previous proposals that this bias is unjustified and that Dpp and Wg act as equally critical and functionally equivalent morphogens in this paradigm [62–64,170,171]. They also argue that both morphogens function equivalently, via the FF mechanism [62–64, 171], rather than acting by distinct, anisotropic mechanisms [170].

## Generality of the feed-forward model

BMP's and Wnts have been implicated in the control of organ growth in diverse animals. Yet, there is no generally accepted explanation for how they exert this control in any single case. Here, we have obtained evidence for 2 central tenets for how Dpp and Wg act to control growth of the *Drosophila* wing. The first is via a progressive expansion in morphogen range, which sustains the continuing increase in cell number and tissue mass within the wing proper. The second is via a reiterative process of recruitment in which morphogen acts together with a short-range signal made by wing cells to induce surrounding, pre-wing cells to grow and enter the wing. We posit that both tenets are fundamental to the dramatic expansion in wing size fueled by Dpp and Wg during larval life.

There is at least one other context in which morphogen spread is coupled to a reiterative process of cell recruitment that controls organ growth, namely the development of the fly eye. In this case, Hedgehog (Hh) made by posteriorly situated photoreceptor cells induces neighboring, anterior cells to express Dpp and both molecules act in concert with additional short range signals to induce more anteriorly situated cells to differentiate into photoreceptors and become new sources of Hh [143–147]. The result is a propagating wave front of Hh and Dpp production and recruitment of new photoreceptor cells into the growing retina.

The FF circuit in the eye differs in several respects from that operating in the wing. Most notably, the ranges of Hh and Dpp expand by the forward propagation of morphogen production rather than the progressive spread from a fixed population of source cells as in the wing. Moreover, photoreceptor recruitment is associated with growth arrest unlike the wing, where the recruited cells continue to proliferate in response to morphogen. Nevertheless, the same central tenets apply: in both cases, growth appears to be governed by a progressive increase in morphogen range coupled to a reiterative process of cell recruitment.

Further study will be required to assess whether, and if so how, these tenets apply in other contexts. For example, Dpp and Wg are both required for growth of the *Drosophila* leg, Dpp from dorsally situated cells and Wg from ventrally situated cells along the A/P compartment boundary. As in the wing, ectopic expression of either morphogen can induce the development of supernumerary legs [10,97,148–150], whereas reduced signaling can result in appendage loss [20,21,151,152]. However, in contrast to the wing, Dpp and Wg appear to exert their control in the leg indirectly via the induction of other signaling molecules [153,154], possibly one or more EGF-like ligands that organize growth as well as patterning along the proximal–distal axis [155,156]. How such ligands act as morphogens to control leg growth remains an open question.

Importantly, challenges to the generality of the FF model apply similarly to most if not all of the models that derive from studies of the wing [29–35]. Given our limited understanding of how morphogens control growth, determining how they do so in any one context, such as the *Drosophila* wing, is an essential first step. In the case of our dissection of the FF mechanism, our findings establish precedents for the roles of morphogen spread, recruitment and FF autoregulation of a selector gene that may well apply in other contexts.

## Materials and methods

### Generation and analysis of mutant clones

In all experiments, we manipulate gene function in genetically marked clones of cells in wing imaginal discs that are mutant for 1 or more of the following genes: *ap*, *dpp*, *wg*, *ft*, *wts*, and/or *d* (as in [64–66]). We then assess the consequences by monitoring the expression of appropriate target genes and/or reporter proteins, particularly in surrounding cells. To generate such clones, we use transgenes in which the promoter (e.g., of the uniformly expressed *Tubulinα1* (*Tub*) gene) is separated from the coding sequence of interest (e.g., *dpp*, *wg* or *vg*) by a "Flp-out" cassette [10,18,97]. These cassettes contain the coding sequence for a given marker protein (e.g., Rat CD2) followed by a transcriptional stop element and in some cases a *yellow+* (*y+*) mini-gene and are flanked by targets for the yeast Flp recombinase (FRTs, denoted by ">"; [16]). Heat shock–induced expression of Flp from a *hsp70.flp* transgene is then used to excise the cassette in each Flp-out transgene (e.g., *Tub>CD2, stop>vg^GFP*), to generate clones of cells that have lost the marker protein and now express the downstream coding sequence (e.g., *Tub>vg^GFP* cells).

We use several variants of this technique that incorporate both Gal4/UAS [94,95] and Gal80 technologies [96,157]. Cassette excision from Flp-out UAS transgenes (e.g., *UAS>CD2, stop>wg<sup>Nrt</sup>*) in the presence of a *Tub>Gal4* "driver" transgene results in clones that express the protein of interest (Wg<sup>Nrt</sup>) and can be monitored either by the loss of expression of a cassette reporter protein (CD2), or more typically, by the gain of expression of the protein of interest. Alternatively, some Flp-out transgenes encode Gal80 inside the cassette (e.g., *Tub>Gal80, stop>vg*), with cassette excision allowing *UAS* transgenes (e.g., *UAS>wg<sup>Nrt</sup>* or *UAS>dpp<sup>GFP</sup>*) to be co-expressed together with the downstream coding sequence (*vg*), under control of Gal4 driver lines that are expressed in the wing disc (e.g., *C765.Gal4* or *rn.Gal4*).

In all experiments, we use standard genetic crosses to generate larvae of the desired genotype (i.e., carrying *hsp70.flp*, as well as all of the requisite Flp-out transgenes, Gal4 drivers, and reporter transgenes), identified as such by the absence of balancer chromosomes carrying the dominant larval marker *Tubby* (*TM6B; SM5-TM6B*). Many of the experimental genotypes contain 2 or more Flp-out transgenes resulting in clones that express different combinations of the transgenes in the same disc (e.g., *UAS>wg<sup>Nrt</sup>, UAS>dpp<sup>GFP</sup>*, and *UAS>wg<sup>Nrt</sup> + UAS>dpp<sup>GFP</sup>*), each identifiable by appropriate markers (e.g., presence of Wg<sup>Nrt</sup>, Dpp<sup>GFP</sup>, or both). We present the key transgenes, as well as the markers used to distinguish the clones, in the figure legends; exact genotypes are provided below.

Except for the experiments described in Fig 10D and 10E, all animals were cultured at 25˚C. Unless otherwise specified in the figure legends (e.g., Fig 11), clones were induced during the first larval instar (24 to 48 hours after egg laying) by heat shock (36˚C or 37˚C, for time spans ranging from 20 minutes to 60 minutes). Wing discs from mature third instar larvae were dissected, fixed, and processed for immunofluorescence by standard methods (e.g., [64–66]) using antisera against native gene products {Vg [64–67]; Sal [158]; Dll [154]; Doc [100]; Ptc [159], epitope tags and reporter proteins [βGalactosidase (Cappell); GFP (native fluorescence); RFP (Molecular Probes or native fluorescence); CD2 (OX34, Serotec), and Phospho-Mad (gift of Ed Laufer and Tom Jessell)}.

For experiments described in Fig 10D and 10E, eggs were collected at 25˚C for 24 hours before being shifted to 20˚C to allow Gal80<sup>ts</sup> to repress *UAS>wg<sup>Nrt</sup>* expression. Larvae were either maintained at 20˚C (control) or shifted to 30˚C for the last 24 or 48 hours of larval life to allow *UAS>wg<sup>Nrt</sup>* expression (experimental).

In all experiments, we focus on the requirements for Dpp, Wg, and the FF signal, or their various transducers, to initiate and sustain "FF wing growth", which we define as the induction and expansion of populations of cells that express Vg under the control of the *vg* QE (see text). We monitor QE activity by assaying for expression of either of 2 QE-reporter transgenes (*1XQE.lacZ* or *5XQE.DSRed*) as well as by the expression of Vg protein in cells in which Notch is not active (as in [64–66]). The results from these experiments were qualitative. For any given genotype, the induction of clones by heat shock either induced FF growth (i.e., QE-dependent Vg expression in abutting cell populations), or it did not. Under the conditions of clone induction, the majority of discs contained multiple clones in the prospective wing domain of each disc, as documented in Figs 3–10. For experimental genotypes used to generate clones, we assayed at least 30 discs (and typically 50 or more such discs) in all cases except for the genotypes shown in Figs 5B and 9C and S2C and S3E Figs, for which we assayed 17, 22, 19, and 19 discs, respectively. For many of these experiments, the distribution of clones also provided internal, negative controls in which some abutting cells received only 1 or 2 of the 3 required inputs and failed to respond in contrast to other abutting cells that received all three (e.g., Figs 5–7; see text). For the experiment depicted in Fig 10D and 10E, we analyzed at least 10 discs from larvae raised under each of the reported experimental conditions.

## Exact genotypes

The following amorphic mutant alleles and transgenes were employed (http://flybase.org/; additional citations below).

**Mutant alleles and duplications.** $ap^{56f}$, $dpp^{d8}$, $dpp^{d10}$, $wg^{GFP}$ [109], $wg^{spd-flg}$, $d^{GC13}$, $ft^{15}$, $vg^{83b27R}$, $wts^{P2}$, M(2)25A.

*Dp(ft+)*: Generated by *hsp70.flp/FRT* mediated recombination between Exelixis transposon insertions d09957 and f03638 [160,161].

**Previously described transgenes and reporters.** $UAS>CD2,y^+>N^*$; $UAS>N^*$. Intact and flipped-out forms of 2 transgenes that encode functionally equivalent, constitutively active forms of Notch ($N^{ECN}$ and $N^{intra}$ [104]; referred to collectively as N* in the text and figure legends, but distinguished in "Experimental Genotypes," below).

$UAS>CD2,y^+>wg^{Nrt}$; $UAS>wg^{Nrt}$; *C765.Gal4* [18].

*C765.Gal4* (active uniformly throughout the wing disc) [18].

$UAS>CD2,y^+>Tkv^{QD}$; $UAS>CD2,y^+>dpp$ [17].

$Tuba1>Gal80,y^+>Gal4$ [64]; *hsp70.flu-GFP* [86]; *rn.Gal4* [162]; $Tuba1.Gal80^{ts}$ [163].

*UAS.ft* [164]; *UAS.yki* [88]; $UAS.d^{RNAi}$ (VDRC #102550); $UAS.ds^{RNAi}$ (VDRC #36219) [106].

*UAS.dpp* and *UAS.wg* (gift of Konrad Basler).

*UAS.FSHβ* [165]; *arm.lacZ* [166]

*5XQE.DsRed* [64]; *1XQE.lacZ* [69]; *omb.lacZ* [99]; *brk.lacZ* ($brk^{X47}$) [48]

**New transgenes.** $UAS>CD2,y^+>dpp^{GFP}$: The coding sequence for $Dpp^{GFP}$ was placed downstream of the UAS promoter [95] but separated from it by a $>CD2,y^+>$ *flp-out* cassette [16]. The $UAS>dpp^{GFP}$ transgene was obtained by excising the *flp-out* cassette in the germline. $Dpp^{GFP}$ contains GFP followed by 3 copies of the flu-tag inserted between amino acids 472 and 477 of Dpp. As a comparison, other published $Dpp^{GFP}$ chimeras have GFP inserted between amino acids 465 and 466 [167] or between amino acids 485 and 486 [168].

$Tuba1>CD2,y^2>vg^{GFP}$: The transgene was assembled by placing a $>CD2, y^2>$ Flp-out cassette that includes the $y^2$ marker downstream of the transcriptional stop [16] between the *Tuba1* promotor [10] and the coding sequence of $vg^{GFP}$ [64].

$Tuba1>Gal80>vg$: The transgene is a derivative of $Tuba1>Gal80,y^+>Gal4$ [64], which contains the coding sequence of *vg* in place of that of Gal4.

$UAS>^mTrap$: A derivative of *UAS.VHH-GFP4::CD8::mCherry* [105] in which the native fluorescence of mCherry was abolished by a K70N mutation [169].

**Experimental genotypes (by figure panel).** **Fig 3. A–C** *y w hsp70.flp/y w hsp70.flp 5XQE. DsRed; $ap^{56f}/dpp^{d10} ap^{56f}$; $UAS>N^{ECN}$; $Tuba1>Gal80,y^+>Gal4/+$.* **D, F** *y w hsp70.flp/y w hsp70.flp 5XQE.DsRed; $dpp^{d8}$ 1xQE-lacZ $ap^{56f}$ /$dpp^{d10} ap^{56f}$ $UAS>N^{ECN}$; $Tuba1>Gal80, y^+>Gal4/+$.* **G** *y w hsp70.flp/y w 5XQE.DsRed; $dpp^{d8} ap^{56f}/dpp^{d10} ap^{56f}$ $UAS>N^{ECN}$; $UAS>dpp^{GFP}/Tuba1>Gal80,y^+>Gal4$.* **H** *y w hsp70.flp/y w hsp70.flp; $dpp^{d8} ap^{56f}/dpp^{d10} ap^{56f}$ $UAS>N^{ECN}$; $UAS>dpp^{GFP}/Tuba1>Gal80,y^+>Gal4$.*

**Fig 4. A** *y w 5XQE.DsRed/y w hsp70.flp 5XQE.DsRed; $dpp^8 ap^{56f}/dpp^{d10} ap^{56f}$; $UAS>CD2, y^+>dpp^{GFP}$ $UAS>N^{intra}/Tuba1>Gal80 >vg$ C765.Gal4.* **B** *y w omb.lacZ/y w hsp70.flp 5XQE. DsRed; $dpp^{d8} ap^{56f}/dpp^{d10} ap^{56f}$; $UAS>CD2,y^+>dpp^{GFP}$ $UAS>N^{intra}/Tuba1>Gal80 >vg$ C765. Gal4.* **C** *y w hsp70.flp/y w hsp70.flp; $dpp^{d8}$ 1XQE-lacZ $ap^{56f}/ dpp^{d10} ap^{56f}$ $UAS> CD2,y^+>N^{ECN}$; $Tuba1>Gal80,y^+>Gal4$ UAS.FSHβ/$UAS>dpp^{GFP}$* (*UAS.FSHβ* [165] serves as a gratuitous control *UAS* transgene in place of the experimental $UAS>^mTrap$ transgene in **4D**). **D** *y w hsp70. flp/y w hsp70.flp; $dpp^{d8}$ 1XQE-lacZ $ap^{56f}/dpp^{d10} ap^{56f}$ $UAS> CD2,y^+>N^{ECN}$; $Tuba1>Gal80, y^+>Gal4$ $UAS>^mTrap/UAS>dpp^{GFP}$.* **E** *y w hsp70.flp/y w hsp70.flp 5XQE.DsRed; $dpp^{d8} ap^{56f}/ dpp^{d10} ap^{56f}$ $UAS> CD2, y^+>N^{ECN}$; 2x $UAS>CD2,y^+>tkv^{QD}$ C765.Gal4/$UAS>^mTrap$.*

**Fig 5. A** *y w hsp70.flp/y w hsp70.flp; dpp$^{d8}$ ap$^{56f}$ UAS>wg$^{Nrt}$/dpp$^{d10}$ ap$^{56f}$; Tubα1>CD2, y$^2$>vg$^{GFP}$ C765.Gal4/UAS.dpp.* **B** *y w hsp70.flp/y w hsp70.flp; dpp$^{d8}$ ap$^{56f}$ UAS>wg$^{Nrt}$/dpp$^{d10}$ ap$^{56f}$; Tubα1>CD2,y$^2$>vg$^{GFP}$ C765.Gal4/MKRS.* **C** *y w hsp70.flp/y w hsp70.flp; dpp$^{d8}$ ap$^{56f}$/ dpp$^{d10}$ ap$^{56f}$; Tubα1>CD2,y$^2$>vg$^{GFP}$ C765.Gal4/UAS.dpp.* **D, E** *y w hsp70.flp/y w hsp70.flp 5XQE.DsRed; dpp$^{d8}$ ap$^{56f}$ UAS>CD2,y$^+$>wg$^{Nrt}$/dpp$^{d10}$ ap$^{56f}$; Tubα1>CD2,y$^2$>vg$^{GFP}$ C765. Gal4/UAS.dpp.* **F** *y w hsp70.flp/y w hsp70.flp 5XQE.DsRed; dpp$^{d8}$ ap$^{56f}$ UAS>wg$^{Nrt}$/dpp$^{d10}$ ap$^{56f}$; Tubα1>CD2,y$^2$>vg$^{GFP}$ C765.Gal4/2xUAS>CD2,y$^+$>tkv$^{QD}$.*

**Fig 6. A, B** *y w hsp70.flp/y w 5XQE.DsRed; dpp$^{d8}$ ap$^{56f}$ UAS>CD2,y$^+$>wg$^{Nrt}$/dpp$^{d10}$ ap$^{56f}$; UAS>dpp$^{GFP}$/ Tubα1>Gal80 >vg C765.Gal4.* **C** *y w hsp70.flp/y w 5XQE.DsRed; dpp$^{d8}$ FRT39 ap$^{56f}$/dpp$^{d10}$ ap$^{56f}$; UAS>CD2,y$^+$>dpp$^{GFP}$ UAS>wg$^{Nrt}$/Tubα1>Gal80 >vg C765.Gal4.* **D** *y w hsp70.flp/y w brk$^{X47}$.lacZ; dpp$^{d8}$ ap$^{56f}$ UAS>CD2,y$^+$>wg$^{Nrt}$/dpp$^{d10}$ ap$^{56f}$; UAS>CD2, y$^+$>dpp$^{GFP}$/Tubα1>Gal80 >vg C765.Gal4.*

**Fig 7. A, B** *y w hsp70.flp/y w hsp70.flp; dpp$^{d8}$ ap$^{56f}$ UAS>CD2,y$^+$>wg$^{Nrt}$/dpp$^{d10}$ ap$^{56f}$; UAS>CD2,y$^+$>dpp$^{GFP}$ UAS.ft/Tubα1>Gal80,y$^+$>Gal4.* **C** *y w hsp70.flp/y w omb.lacZ; dpp$^{d8}$ ap$^{56f}$/dpp$^{d10}$ ap$^{56f}$; UAS>CD2,y$^+$>dpp$^{GFP}$ UAS.ft/Tubα1>Gal80,y$^+$>Gal4 UAS.wg.* **D** *y w hsp70.flp/y w hsp70.flp 5XQE.DsRed; dpp$^{d8}$ ap$^{56f}$/dpp$^{d10}$ UAS.d$^{RNAi}$ ap$^{56f}$; UAS>CD2, y$^+$>dpp$^{GFP}$ UAS.ft UAS>wg$^{Nrt}$/Tubα1>Gal80,y$^+$>Gal4.*

**Fig 8. A** *y w hsp70.flp/y w hsp70.flp 5XQE.DsRed; dpp$^{d8}$ ap$^{56f}$/dpp$^{10}$ ap$^{56f}$; UAS>dpp$^{GFP}$ UAS>wg$^{Nrt}$/Tubα1>CD2,y$^2$>vg$^{GFP}$ C765.Gal4.* **B** *y w hsp70.flp/y w hsp70.flp 5XQE.DsRed; dpp$^{d8}$ d$^{GC13}$ FRT39 ap$^{56f}$ fj.lacZ/dpp$^{d10}$ d$^{GC13}$ FRT39 ap$^{56f}$; UAS>dpp$^{GFP}$ UAS>wg$^{Nrt}$/ Tubα1>CD2,y$^2$>vg$^{GFP}$ C765.Gal4.* **C** *y w hsp70.flp/y w hsp70.flp 5XQE.DsRed; dpp$^{d8}$ hsp70. GFP$^{HA}$ FRT39 ap$^{56f}$/dpp$^{d10}$ d$^{GC13}$ FRT39 ap$^{56f}$; UAS.dpp UAS>wg$^{Nrt}$/Tubα1>CD2,y$^2$>vg$^{GFP}$ C765.Gal4.* **D** *y w hsp70.flp/y w hsp70.flp; dpp$^{d8}$ hsp70.GFP$^{HA}$ FRT39 ap$^{56f}$/dpp$^{d10}$ d$^{GC13}$ FRT39 ap$^{56f}$; UAS.dpp UAS>wg$^{Nrt}$/Tubα1>CD2,y$^2$>vg$^{GFP}$ C765.Gal4.* **E, F** *y w hsp70.flp/y w hsp70.flp 5XQE.DsRed; dpp$^{d8}$ hsp70.GFP$^{HA}$ M(2)25A FRT39 ap$^{56f}$/dpp$^{d10}$ d$^{GC13}$ FRT39 ap$^{56f}$; UAS.dpp UAS>wg$^{Nrt}$/Tubα1>CD2,y$^2$>vg$^{GFP}$ C765.Gal4.*

**Fig 9. A** *y w hsp70.flp 5XQE.DsRed/y w hsp70.flp 5XQE.DsRed; dpp$^{d10}$ ft$^{15}$ FRT39 ap$^{56f}$/ dpp$^{d10}$ ft$^{15}$ FRT39 ap$^{56f}$; C765.Gal4/C765.Gal4.* **B, C** *y w hsp70.flp/y w hsp70.flp 5XQE.DsRed; dpp$^{d10}$ ft$^{15}$ FRT39 ap$^{56f}$/dpp$^{d10}$ ft$^{15}$ FRT39 ap$^{56f}$; UAS>CD2,y$^+$>wg$^{Nrt}$ C765.Gal4/UAS.dpp.* **D** *y w hsp70.flp/y w hsp70.flp 5XQE.DsRed; dpp$^{d10}$ ft$^{15}$ FRT39 ap$^{56f}$/dpp$^{d10}$ ft$^{15}$ FRT39 ap$^{56f}$ UAS> wg$^{Nrt}$; 2x UAS>CD2,y$^+$>tkv$^{QD}$ C765.Gal4/MKRS.* **E** *y w hsp70.flp/y w 5XQE.DsRed; dpp$^{d10}$ ft$^{15}$ FRT39 ap$^{56f}$/dpp$^{d8}$ ft$^{15}$ FRT39 ap$^{56f}$; UAS>CD2,y$^+$>dpp$^{GFP}$ UAS>wg$^{Nrt}$/C765.Gal4.* **F** *y w hsp70.flp/y w 5XQE.DsRed; dpp$^{d10}$ ft$^{15}$ FRT39 ap$^{56f}$/dpp$^{d10}$ ft$^{15}$ FRT39 ap$^{56f}$; UAS>CD2, y$^+$>wg$^{Nrt}$ C765.Gal4/UAS>CD2,y$^+$>dpp$^{GFP}$.* **G** *y w hsp70.flp/y w omb.lacZ 5XQE.DsRed; dpp$^{d10}$ ft$^{15}$ FRT39 ap$^{56f}$/dpp$^{d10}$ ft$^{15}$ FRT39 ap$^{56f}$; UAS>CD2,y$^+$>wg$^{Nrt}$ C765.Gal4/UAS>CD2, y$^+$>dpp$^{GFP}$.*

**Fig 10. A** *y w hsp70.flp/y w hsp70.flp 5XQE.DsRed; dpp$^{d8}$ ap$^{56f}$ UAS>CD2,y$^+$>wg$^{Nrt}$/dpp$^{d10}$ ap$^{56f}$; Tubα1>CD2,y$^2$>vg$^{GFP}$ C765.Gal4/UAS.dpp.* **B, C** *y w hsp70.flp/y w hsp70.flp 5XQE. DsRed; dpp$^{d8}$ wg$^{spd}$ ap$^{56f}$ UAS>CD2,y$^+$>wg$^{Nrt}$/dpp$^{d10}$ wg$^{spd}$ ap$^{56f}$; Tubα1>CD2,y$^2$>vg$^{GFP}$ C765. Gal4/UAS.dpp.* **D** *y w hsp70.flp/y w hsp70.flp 5XQE.DsRed; ft$^{15}$ ap$^{56f}$/ft$^{15}$ wg$^{spd}$ ap$^{56f}$ Tubα1. Gal80$^{ts}$; UAS> wg$^{Nrt}$; C765.Gal4.* **E** *y w hsp70.flp/y w hsp70.flp 5XQE.DsRed; ft$^{15}$ wg$^{spd}$ ap$^{56f}$/ft$^{15}$ wg$^{spd}$ ap$^{56f}$ Tubα1.Gal80$^{ts}$; UAS>wg$^{Nrt}$/C765.Gal4.* **F** *y w hsp70.flp/y w hsp70.flp; ft$^{15}$ wg$^{GFP}$ ap$^{56f}$/dpp$^{d10}$ ft$^{15}$ FRT39 ap$^{56f}$; Tubα1>Gal80,y$^+$>Gal4/rn.Gal4 UAS>$^m$Trap.*

**Fig 11. A, B** *y w hsp70.flp/y w hsp70.flp 5XQE.DsRed; tkv$^{a12}$ FRT40/arm.lacZ FRT40; Tubα1>CD2,y$^2$>vg$^{GFP}$.* **C** *y w hsp70.flp/y w hsp70.flp; FRT42 vg$^{83b27R}$/FRT42 Tubα1.DsRed; Tubα1>CD2,y$^2$>vg$^{GFP}$/1XQE.lacZ.*

## Supporting information

**S1 Fig. Induction of FF growth by Wg, Dpp, and the FF signal is restricted to the pre-wing (Tsh$^{OFF}$ Hth$^{OFF}$) region of the wing disc. (A, B)** *Tub>vg$^{GFP}$* clones (marked by GFP, bright yellow in the micrographs) in *dpp$^d$ ap$^o$ UAS.dpp UAS>wg$^{Nrt}$* discs monitored for either Vg (purple, **A**) or the *5XQE.DsRed* reporter (red, **B**), as well as Hth (turquoise, **A** and **B**), a transcription factor that is expressed in prospective hinge and body wall territories surrounding the pre-wing domain (as shown for the corresponding *wild-type* discs to the right). The clones have induced FF growth in the surround (indicated by the halos of Vg and *5XQE.DsRed* expressing cells); however, this is restricted to the pre-wing domain delimited by the surrounding Hth expressing hinge and body wall territories. **(C)** A *dpp$^d$ ap$^o$ UAS.dpp* disc containing a *Tub>vg$^{GFP}$ UAS>wg$^{Nrt}$* clone (bright yellow) next to a *UAS>wg$^{Nrt}$* (dim yellow) monitored for Wg, Vg$^{GFP}$, and *5XQE.DsRed* (labeled as in **B**), as well as the transcription factor Tsh (turquois), which like Hth, demarcates the proximal hinge and body wall territories of the disc (as in the *wild-type* disc to the right). The *Tub>vg$^{GFP}$ UAS>wg$^{Nrt}$* clone expresses the *5XQE. DsRed* reporter and has induced abutting *wild-type* cells to do the same (confirming that this clone is *UAS>wg$^{Nrt}$* in addition to *Tub>vg$^{GFP}$*). This clone has also induced extensive FF growth throughout the abutting *UAS>wg$^{Nrt}$* clone and its immediate *wild-type* neighbors (monitored by *5XQE.DsRed* expression). As in (**A, B**), all of the FF growth has occurred within the pre-wing region of the disc surrounded by the Tsh$^{ON}$ Hth$^{ON}$ hinge and body wall territories. **Key genotypes:** *dpp$^d$ ap$^o$ Tub>stop>vg$^{GFP}$ C765.Gal4* discs that are also *UAS.dpp UAS>wg$^{Nrt}$* (**A, B**), or that are *UAS.dpp UAS>stop>wg$^{Nrt}$* (**C**). **Exact genotypes: A, B** *y w hsp70.flp/y w hsp70.flp; dpp$^{d8}$ ap$^{56f}$ UAS>wg$^{Nrt}$/dpp$^{d10}$ ap$^{56f}$; Tubα1>CD2,y$^2$>vg$^{GFP}$ C765. Gal4/UAS.dpp.* **C** *y w hsp70.flp/y w hsp70.flp 5XQE.DsRed; dpp$^{d8}$ ap$^{56f}$ UAS>CD2,y$^+$>wg$^{Nrt}$/ dpp$^{d10}$ ap$^{56f}$; Tubα1>CD2,y$^2$>vg$^{GFP}$ C765.Gal4/UAS.dpp.* Dpp, Decapentaplegic; FF, feed-forward; FRT, >, Flp recombinase targets; Hth, Homothorax; QE, Quadrant enhancer; Tsh, Teashirt; *Tub, Tubulinα1*; Vg, Vestigial; Wg, Wingless; Wg$^{Nrt}$, Neurotactin-Wingless. (TIF)

**S2 Fig. Dpp is required together with Wg to induce FF growth in *wts$^o$* discs. (A–D)** *dpp$^d$ ap$^o$ wts$^o$* discs carrying either *UAS>dpp$^{GFP}$* (**A**, green), *UAS>wg$^{Nrt}$ UAS>dpp$^{GFP}$* (**B**, green/ blue) or abutting *UAS>dpp$^{GFP}$* and *UAS>wg$^{Nrt}$* clones (**C, D**, green and blue, respectively), monitored for *5XQE.DsRed* expression (**A–C,** red) or native Vg (**D**, purple). Uniform activity of the FF transduction pathway is indicated by yellow wash in the diagrams to the left; the Wg IR is indicated by yellow asterisks (**A–D**) and a dotted blue line (**B, C**) in the micrographs to the right. In *dpp$^d$ ap$^o$ wts$^o$* discs, as in *dpp$^d$ ft$^o$ ap$^o$* discs (Fig 9A), *5XQE.DsRed* activity is not detected and no FF growth occurs, even though the FF transduction pathway is constitutively active. **(A)** *UAS>dpp$^{GFP}$* clones generated in this background restore weakly detectable QE activity (red) and an expansion of the pre-wing territory, likely dependent on cryptic Wg expressed by IR cells in the prospective hinge (as in Figs 9B and 10D; note that the level of *5XQE.DsRed* is elevated modestly in cells neighboring the IR). **(B)** *UAS>dpp$^{GFP}$ UAS>wg$^{Nrt}$* clones (outlined in white) restore peak QE-dependent activity within the clone and abutting cells outside the clone and grow as wing tissue. **(C, D)** Abutting *UAS>dpp$^{GFP}$* and *UAS>wg$^{Nrt}$* clones in *dpp$^d$ ap$^o$ wts$^o$* discs (Fig 9F and 9G). As in *dpp$^d$ ft$^o$ ap$^o$* discs (Fig 9F and 9G), cells within the *UAS>wg$^{Nrt}$* clone (outlined in white in **C** and in the inset in **D**), as well as their immediate neighbors, express peak levels of both *5XQE.DsRed* (**C**) and native Vg (**D**) and grow as wing tissue, provided that they are located close enough to the abutting *UAS>dpp$^{GFP}$* clones to receive Dpp. **Key genotypes:** *dpp$^d$ ap$^o$ wts$^o$ C765.Gal4 UAS>stop>dpp$^{GFP}$ UAS>stop>wg$^{Nrt}$* discs (**A–D** all have *UAS>dpp$^{GFP}$* clones, whereas **B, C**, but not **A**, have *UAS>wg$^{Nrt}$* clones). **Exact genotypes: A–C** *y w hsp70.flp/y w hsp70.flp 5XQE.DsRed; dpp$^{d8}$ ap$^{56f}$ UAS>CD2,*

$y^+>wg^{Nrt}/dpp^{d10}\ ap^{56f};\ wts^{P2}\ UAS>CD2,y^+>dpp^{GFP/}wts^{P2}\ C765.Gal4$ **D** $y\ w\ hsp70.flp/y\ w$ $hsp70.flp;\ dpp^{d8}\ ap^{56f}\ UAS>CD2,y^+>wg^{Nrt}/dpp^{d10}\ ap^{56f};\ wts^{P2}\ UAS>CD2,y^+>dpp^{GFP/}wts^{P2}$ $C765.Gal4.$ *ap*, apterous; Dpp, Decapentaplegic; FF, feed-forward; FRT, >, Flp recombinase targets; Ft, Fat; IR, inner ring; QE, Quadrant enhancer; Vg, Vestigial; Wg, Wingless; Wg^Nrt, Neurotactin-Wingless; Wts, Warts.
(TIF)

**S3 Fig. Dpp is required together with Wg to induce FF growth in Yki over-expressing discs.** (**A–D**) $dpp^d\ ap^o\ UAS.yki$ discs carrying either a single $UAS>dpp^{GFP}$ clone (**A**), or abutting $UAS>wg^{Nrt}$ and $UAS>dpp^{GFP}$ clones (**B–D**), assayed for *5XQE.DsRed* (red, **A** and **B**) or native Vg expression (purple, **C** and **D**). $UAS>dpp^{GFP}$ clones are visualized by native fluorescence of the GFP tag (green) while the $UAS>wg^{Nrt}$ clones are marked either directly (**B**, **C**) by Wg^Nrt expression (blue) or indirectly (**D**) through their ability to induce peak Vg levels (purple) autonomously and at short-range (as in **C**). (**A**) $UAS>dpp^{GFP}$ clones in $dpp^d\ ap^o\ UAS.yki$ discs restore barely detectable QE activity and an expansion of the pre-wing territory (as in S2A Fig, *5XQE.DsRed* expression is weakly elevated in pre-wing cells neighboring the surrounding Wg IR, consistent with IR cells being the source of cryptic Wg responsible for the QE response). (**B–D**) In contrast, $UAS>dpp^{GFP}$ clones induce high-level QE-dependent Vg expression and extensive growth of neighboring $UAS>wg^{Nrt}$ clones and their immediate neighbors, provided that the cells are close enough to receive Dpp (confirmed in **D** by expression of the *omb.lacZ* reporter; green, right panel; the $UAS>wg^{Nrt}$ clone border is indicated by a white line in the inset in (**C**). (**E**) A $dpp^d\ ap^o\ UAS.yki$ disc carrying $UAS>tkv^{QD}$ clones (marked by induction of *omb.lacZ* expression, green), $UAS>wg^{Nrt}$ clones (marked by Wg^Nrt expression, blue) and $UAS>tkv^{QD}\ UAS>wg^{Nrt}$ clones (green/blue). Vg expression (purple) is strictly autonomous to the $UAS>tkv^{QD}\ UAS>wg^{Nrt}$ clones. **Key genotypes:** $dpp^d\ ap^o\ UAS.yki\ C765.$ $Gal4\ UAS>stop>wg^{Nrt}$ discs with either $UAS>stop>dpp^{GFP}$ (**A–D**) or $UAS>stop>tkv^{QD}$ (**E**). **Exact genotypes: A, B** $y\ w\ hsp70.flp/y\ w\ 5XQE.DsRed;\ dpp^{d8}\ ap^{56f}/dpp^{d10}\ ap^{56f};\ UAS>CD2,$ $y^+>dpp^{GFP}\ UAS.yki;\ UAS>CD2,y^+>wg^{Nrt}\ C765.Gal4.$ **C, D** $y\ w\ hsp70.flp/y\ w\ omb.lacZ;\ dpp^{d8}$ $ap^{56f}/dpp^{d10}\ ap^{56f};\ UAS>CD2,y^+>dpp^{GFP}\ UAS.yki;\ UAS>CD2,y^+>wg^{Nrt}\ C765.Gal4.$ **E** $y\ w$ $hsp70.flp/y\ w\ omb.lacZ;\ dpp^{d8}\ ap^{56f}\ UAS>CD2,y^+>wg^{Nrt}/dpp^{d10}\ FRT39\ ap^{56f};\ 2xUAS>CD2,$ $y^+>tkv^{QD}\ C765.Gal4/UAS.yki.$ *ap*, apterous; Dpp, Decapentaplegic; FF, feed-forward; FRT, >, Flp recombinase targets; IR, inner ring; QE, Quadrant enhancer; Tkv, Thickveins; Vg, Vestigial; Wg, Wingless; Wg^Nrt, Neurotactin-Wingless; Yki, Yorkie.
(TIF)

**S4 Fig. Ft is the FF signal: Cell-autonomous vs. nonautonomous activation of FF growth by** $ft^o$ **vs.** $UAS.ds^{RNAi}$ **clones in** $dpp^d\ ap^o\ UAS>wg^{Nrt}\ UAS>dpp$ **discs.** (**A**) A $dpp^d\ ap^o$ disc with uniform Wg^Nrt expression that contains 2 large clones both of which are mutant for *ft* (yellow in the cartoon; marked "black" by the loss of an *hsp70.GFP* marker, yellow, in the middle and right most micrographs), and one of which ectopically expresses Dpp (outlined in green; and marked "black" by the loss of CD2, green, resulting from excision of the stop cassette in $UAS>CD2,stop>dpp$). Cells within both clones grow autonomously as *5XQE.DsRed* expressing wing tissue. Importantly, QE activation is strictly confined to the $ft^o$ clones, indicating that *vg*-expressing wing cells that lack Ft cannot induce FF growth of abutting *wild-type* cells (X's in the cartoon), even when the latter receive both Dpp and Wg^Nrt (a third $ft^o$ clone located to the left of these 2 $ft^o$ clones fails to express *5XQE.DsRed* because it is located outside of the pre-wing territory). (**B**) A $dpp^d\ ap^o$ disc containing 2 $UAS>wg^{Nrt}\ UAS.ds^{RNAi}$ clones (blue, marked by Wg^Nrt expression), one of which expresses Dpp^GFP (green). Cells within these clones, as well as surrounding cells that can receive the tethered Wg^Nrt signal, express *5XQE.DsRed* and grow as wing tissue, provided that they are positioned close enough to the Dpp^GFP expressing

clone (cells that we infer are out of range are indicated by X's). Thus, loss of either *ft* or *ds* activity cell-autonomously activates QE-dependent Vg expression in $dpp^d$ $ap^o$ discs that are supplied with exogenous Dpp and Wg, but only clones that have lost *ds* activity can induce FF growth of the surround, whereas those that have lost *ft* activity cannot, indicating that Ft itself is the *vg*-dependent, *vg*-inducing FF signal. **Key genotypes: (A)** $dpp^d$ $ap^o$ $UAS{>}wg^{Nrt}$ $UAS{>}CD2,stop{>}dpp$ *C765.Gal4* disc that is also $ft^0$ FRT39/*hsp70.GFP* FRT39. **(B)** $dpp^d$ $ap^o$ *UAS.ds$^{RNAi}$ UAS>wg$^{Nrt}$ UAS>CD2,stop>dpp$^{GFP}$ Tub>Gal80,stop>Gal4* disc. **Exact genotypes: A** *y w hsp70.flp/y w hsp70.flp 5XQE.DsRed; dpp$^{d8}$ ft$^{15}$ FRT39 ap$^{56f}$/dpp$^{d10}$ Dp(ft$^+$) hsp70. GFP$^{HA}$ FRT39 ap$^{56f}$; UAS>CD2,y$^+$>dpp UAS>wg$^{Nrt}$/C765.Gal4.* **B** *y w hsp70.flp/y w hsp70.flp 5XQE.DsRed; dpp$^{d10}$ ap$^{56f}$/dpp$^{d8}$ UAS.ds$^{RNAi}$ ap$^{56f}$; UAS>CD2,y$^+$>dpp$^{GFP}$ UAS>wg$^{Nrt}$/ Tubα1>Gal80,y$^+$>Gal4. ap,* apterous; Dpp, Decapentaplegic; Ds, Dachsous; FF, feed-forward; FRT, >, Flp recombinase targets; Ft, Fat; QE, Quadrant enhancer; Vg, Vestigial; Wg, Wingless; Wg$^{Nrt}$, Neurotactin-Wingless.
(TIF)

## Acknowledgments

We thank Andrew Tomlinson, Rory Coleman, Joe Parker, and Paul Langridge for advice, discussion, and comments on the MS, Ingolf Reim, Sean Carroll, Richard Mann, Rosa Barrio, Jose Felix de Celis, Ed Laufer, Tom Jessell, and the Developmental Studies Hybridoma Bank for antisera and Konrad Basler for transgenes.

## Author Contributions

**Conceptualization:** Myriam Zecca, Gary Struhl.

**Data curation:** Myriam Zecca.

**Formal analysis:** Myriam Zecca.

**Funding acquisition:** Gary Struhl.

**Investigation:** Gary Struhl.

**Methodology:** Myriam Zecca, Gary Struhl.

**Project administration:** Myriam Zecca, Gary Struhl.

**Resources:** Gary Struhl.

**Supervision:** Gary Struhl.

**Validation:** Myriam Zecca.

**Visualization:** Myriam Zecca, Gary Struhl.

**Writing – original draft:** Myriam Zecca, Gary Struhl.

**Writing – review & editing:** Myriam Zecca, Gary Struhl.

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
