## [Editor Report · Decision Letter 0]

17 Jul 2020

Dear Gary, 

Thank you for submitting your manuscript entitled "A unified mechanism for the control of Drosophila wing growth by the morphogens Decapentaplegic and Wingless" for consideration as a Research Article by PLOS Biology. Thank you also for your patience as we completed our editorial process, and please accept my apologies for the delay in providing you with our decision.

Your manuscript has now been evaluated by the PLOS Biology editorial staff as well as by an academic editor with relevant expertise and I am writing to let you know that we would like to send your submission out for external peer review.

Please re-submit your manuscript within two working days, i.e. by Jul 20 2020 11:59PM.

Kind regards,

Ines

--

Ines Alvarez-Garcia, PhD

Senior Editor

PLOS Biology

---

## [Decision Letter · Decision Letter 1]

14 Sep 2020

Dear Gary,

Thank you very much for submitting your manuscript "A unified mechanism for the control of Drosophila wing growth by the morphogens Decapentaplegic and Wingless" for consideration as a Research Article at PLOS Biology. Thank you also for your patience as we completed our editorial process, and please accept my sincere apologies for the delay in providing you with our decision. Your manuscript has been evaluated by the PLOS Biology editors, an Academic Editor with relevant expertise, and by several independent reviewers.

You will see that the reviewers are all impressed with the elegant and sophisticated experiments performed in the manuscript and in general agree with the role of dpp in the feedforward model in a very similar manner than wg. However, they also have mixed feelings regarding the general advance of the conclusions for the field. Nevertheless, both Reviewers 1 and 2 think that unifying the results contributes to clarify the role of these pathways in wing growth and that there is enough merit to consider the manuscript for publication. After discussing the reviews with the editorial team and consider the potential advance, we have decided to invite you to submit a revision. Such revision should address all the points raised by Reviewer 2, including the suggestion to move some of the main figures to the supplementary files as this would help to improve the flow of the paper.

In light of the reviews (attached below), we will not be able to accept the current version of the manuscript, but we would welcome re-submission of a much-revised version that takes into account the reviewers' comments. We cannot make any decision about publication until we have seen the revised manuscript and your response to the reviewers' comments. Your revised manuscript is also likely to be sent for further evaluation by the reviewers.

We expect to receive your revised manuscript within 3 months. 

**IMPORTANT - SUBMITTING YOUR REVISION**

*Re-submission Checklist*

*Published Peer Review*

*PLOS Data Policy*

*Blot and Gel Data Policy*

Sincerely,

Ines

--

Ines Alvarez-Garcia, PhD,

Senior Editor,

ialvarez-garcia@plos.org,

PLOS Biology

Reviewers’ comments

Rev. 1:

In this manuscript Zecca and Struhl analyze the contribution of Wg, Dpp and Fat/Ds signaling to regulate vg expression and consequently the generation and expansion of the wing anlage of the Drosophila wing imaginal disc. They generate clones of cells expression a variety of genes in double mutant "ap dpp" wing discs, a genotype that precludes the generation of the wing primordium. In these discs they could unambiguously determine the individual and combined requirements for Dpp, Wg and Fat/Ds by providing the expression of these genes through the excision of transcriptional stop cassettes. In these discs they monitored the expression of several reporter genes, and use the non-autonomous expression of the vg quadrant enhancer as indication of the presence of wing tissue specified by induced vg expression. The experimental design is extremely sophisticated, but the authors have made a great effort to present their results in a way that could be followed by readers not familiar with the intricacies of clonal analysis and wing disc development. The manuscript is written in a way that the experimental and conceptual logic can be followed effortless, and the extensive use of colorful drawings and consistent color codes really help to visualize and understand the results. I understand that part of the analysis of conclusions related to Wg were already published, but I believe that putting together the results for Wg, Dpp and Wg+Dpp makes all sense and contributes to clarify the combined contribution of these pathways to wing growth. I can only congratulate the authors for yet another excellent piece of work and are in favor of publishing this manuscript as it is.

Rev. 2:

Zecca and Struhl investigate growth control in the wing imaginal disc of Drosophila, specifically how the morphogens Wingless (Wg) and Decapentaplegic (Dpp) function together with the protocadherins Fat (Ft) and Dachsous (Ds) regulate this. Here they extend their previous studies, integrating Dpp into their 'feedforward' model. The model explains how Wg and Dpp can expand the size of the wing primordium in collaboration with the protocadherins Fat, so its size is limited by the distance to which these morphogens can move within the the wing disc. In addition, it is shown that both morphogens are continually required within the wing to promote proliferation. The simple proposal is that proliferation just requires a low threshold of both Wg and Dpp and above that the rate is constant. This is a much more attractive model that others that have been proposed, not least because it can easily explain why proliferation is fairly uniform throughout the developing wing pouch.

A wealth of data is presented to support the model, most of it elegant and well explained. The figure legends are detailed and although there is some duplication with the main text this is very useful as the figures are quite complicated. The cartoons provided adjacent the images are also very helpful. The discussion is generally pertinent, although some issues are still not completely explained away, such as whether there are differences in growth control between lateral and medial regions of the wing, and in particular why membrane tethered Wg can drive such high levels of growth. The latter issue has probably hampered acceptance of the author's previous studies, but now with the addition of Dpp into all this, with Dpp generally being accepted and a moveable morphogen in the wing, the model presented should gain wider appeal. This is a significant paper and all previous work should be reinterpreted in light of the data presented here.

Other comments

The number of figures, however, cannot be ignored…are they all necessary? Most appear essential, but the authors could go through them again and decide what could be moved to Supplementary Data. Some suggestions: Dpp morpho trap experiments (the tkvQ is a better reagent), all the Fig. 10 experiments

Interpretation of sal expression can be a little difficult given its widespread expression outside of the wing pouch (future experiments might consider using some of the reporters that drive only in the wing pouch).

Fig. 5E sal expression in the wing pouch is apparently not dependent on vg (Barrio and De Celis (2004) PNAS)…as well as suggesting it can't be assumed that sal cells also express vg-QE, it also asks the question of whether loss of vg results in complete loss of wing identity

The premiss that Ft is a ligand and Ds is a receptor and is directly activating D downstream is not 'standard' theory in growth control (which has Ds as the ligand, Ft as the receptor which blocks D activity), although there is limited evidence to support this now and this could be pointed out, as some readers will be confused.

Relevance to other tissues? What other examples are there of tissue specific transcription factors being regulated by morphogens throughout the whole tissue. Is the model relevant only to small tissues such as imaginal discs or very early embryos, what happens when tissues get too big for morphogens to reach all the cells (and the tissue needs to get bigger)?

When does feed forward end, or does it ever end before pupation?

No scale bars, it is difficult to determine how much growth has occurred compared to a real wing

4C cartoon does not match the real image - it does in the other images so was a little confusing to start with, and I don't really see the Dpp-GFP in the N* clone.

It would have helped to have some of these discs stained for Tsh as well, just to be sure that anywhere vgQE is not active is not simply because a clone abuts cells that can never express it (eg regions labeled x in Fig. 4)

Vg and Yki are both coactivators….and both bind the same TF, Scalloped…is this just a coincidence?

Further…it would have been useful to stain the discs for scalloped expression that are also constitutively expressing vg and which also tkv or arrow mutant clones to see if Wg and Dpp are driving scalloped expression

Rev. 3:

The Drosophila wing is an excellent system to study the mechanisms that couple organ identity and growth, a key and long-standing problem in developmental biology which remains poorly understood. During wing development in the fly, the vestigial (vg) gene marks its wing primordium and is necessary for its development. The expression of vg is controlled by a number of enhancers, including early initiating ones (vgBE, vg boundary enhancer) and, at least, a later autoregulatory enhancer, vgQE (vg quadrant enhancer). Work since the 90's has identified major regulatory inputs on vg coupled to patterning events, such as the early dorso-ventral subdivision of the wing disc.

Work by the authors of this manuscript have shown that, after a priming of the prospective wing by early vg expression, the vgQE is necessary and sufficient to provide for enough vg activity to rescue wing development (in apterous mutants. What the contribution of vgQE to the development of the adult wing is still awaits studies where this enhancer is specifically deleted from its genomic locus). After defining a series of upstream regulators, that include the (dorso-ventral, DV) activation of Notch and Wg/Wnt signaling, the authors went on to describe a mechanism that explains the coupling of wing growth and wing-fate assignment in the growing tissue. This mechanism is based on a feedforward (FF) loop capable of recruiting pre-wing cells non-autonomously provided that these recruited cells receive Wg signal while, simultaneously, drives their growth through the repression of the Hippo signaling cascade. In all this mechanism, the recruitment of the pre-wing cells rests on the activation of the vgQE (within the range of Wg's reach) -and therefore, vgQE regulation is critical. In addition to this string of papers, dpp (decapentaplegic, a BMP2 family gene) and its signaling pathway has been shown to play an essential role in wing development. dpp is expressed orthogonally to the DV boundary signaling center and is essential to support the growth of the wing disc and to pattern gene expression along the AP axis. In the present manuscript, the authors ask whether dpp is also required for the FF recruitment and growth of the wing primordium. The authors go a long way, using sophisticated genetic combinations, to show that dpp acts very much like wg -that is, it is required (together with Wg) for the non-autonomous expansion of the vgQE-driven expression of vg. However, as much impressed I am by the genetic analysis and convinced by the FF model, the results presented here were hardly unexpected. It was shown by Kim and co-workers (Nature 1997) that the vgQE is directly regulated by the Dpp signaling pathway by binding of P-Mad (its signal transducer) to (at least) one specific binding site. Therefore, the FF ought to depend on dpp signaling. The fact that Dpp acts as a morphogen (or at least, its effective diffusion range is limited) would naturally limit the region within the disc that is under Dpp's influence. Less critical, but equally important, Dpp signaling has been shown to also modulate the Hippo/yki pathway as a way to stimulate tissue growth in the same direction as the FF recruitment model (Rogulja et al., Dev Cell 2008), so that, in addition to the vg-regulated ft/Ds interface that feeds on the Hippo/yki pathway, dpp signaling might contribute as well. Clones that are defective for the Dpp signaling are known to grow poorly in the wing pouch, so that one would expect that, in rescued wings, the same would hold. A less expected finding is that, in the absence of a wing pouch Wg source, the wing surrounding source of Wg (Wg inner ring, IR) can provide for some Wg and activation of the vgQE. In mutants that lack this expression (spadeflag) the wing hinge is affected, but the adult wing blade develops. Analysis of the spadeflag wings quantitatively may tell whether the IR contribution is at all relevant for wing development if the wg pouch is present. Frankly, I think that the paper does not constitute a sufficiently large and original improvement of the FF mechanism that the authors have put forward in the past.

---

## [Decision Letter · Decision Letter 2]

22 Jan 2021

Dear Gary,

Thank you for submitting the revised version of your manuscript entitled "A unified mechanism for the control of Drosophila wing growth by the morphogens Decapentaplegic and Wingless" for publication in PLOS Biology as a Research Article.

The PLOS Biology editors have seen your revision, along with the Academic Editor, Nicolas Tapon, and one of the original reviewers. I am pleased to say that we can in principle offer to publish your manuscript in PLOS Biology, provided you address any remaining formatting and reporting issues. These will be detailed in an email that will follow this letter and that you will usually receive within 2-3 business days, during which time no action is required from you. Please note that we will not be able to formally accept your manuscript and schedule it for publication until you have made the required changes.

PRESS

Thank you again for supporting Open Access publishing. We look forward to publishing your paper in PLOS Biology. 

Sincerely, 

Ines

--

Ines Alvarez-Garcia, PhD 

Senior Editor 

PLOS Biology

Reviewers’ comments

Rev. 2:

The authors have addressed all my issues (and those of the other reviewers) and so I recommend publication.